# Memory effects on greenhouse gas emissions ($CO_2$, $N_2O$ and $CH_4$) following
# grassland restoration?
Lutz Merbold[1,2*], Charlotte Decock[3+], Werner Eugster[1], Kathrin Fuchs[4], Benjamin Wolf[4], Nina
Buchmann[1] and Lukas Hörtnagl[1]
[1] Department of Environmental Systems Science, Institute of Agricultural Sciences, Grassland
Sciences Group, ETH Zurich, Universitätsstrasse 2, 8092 Zürich, Switzerland
[2] Mazingira Centre, International Livestock Research Institute (ILRI), Old Naivasha Road, PO
Box 30709, 00100 Nairobi, Kenya
[3] Department of Environmental Systems Science, Institute of Agricultural Sciences,
Sustainable Agro-ecosystem Group, ETH Zurich, Universitätsstrasse 2, 8092 Zürich,
Switzerland, [+] now at: Department of Natural Resources Management and Environmental
Sciences, California State University, San Luis Obispo, California, USA
[4] Institute for Meteorology and Climate Research (IMK-IFU), Karlsruhe Institute of
Technology (KIT), Kreuzeckbahnstrasse 19, 82467 Garmisch-Partenkirchen, Germany
[*] corresponding author: lutz.merbold@gmail.com
**Keywords:** eddy covariance, global warming potential, manual static chamber, management,
background greenhouse gas emissions, ploughing, fertilization
## Abstract
A five-year greenhouse gas (GHG) exchange study of the three major gas species ($CO_2$, $CH_4$
and $N_2O$) from an intensively managed permanent grassland in Switzerland is presented.
Measurements comprise two years (2010/2011) of manual static chamber measurements of
$CH_4$ and $N_2O$, five years of continuous eddy covariance (EC) measurements ($CO_2$/$H_2O$ – 2010-
2014) and three years (2012-2014) of EC measurement of $CH_4$ and $N_2O$. Intensive grassland
management included both regular and sporadic management activities. Regular management
practices encompassed mowing (3-5 cuts per year) with subsequent organic fertilizer
amendments and occasional grazing whereas sporadic management activities comprised
grazing or similar activities. The primary objective of our measurements was to compare pre-
ploughing to post-ploughing GHG exchange and to identify potential memory effects of such
a substantial disturbance on GHG exchange and carbon (C) and nitrogen (N) gains/losses. In
order to include measurements carried out with different observation techniques, we tested two
different measurement techniques jointly in 2013, namely the manual static chamber approach
and the eddy covariance technique for $N_2O$, to quantify the GHG exchange from the observed
grassland site.
Our results showed that there were no memory effects on $N_2O$ and $CH_4$ emissions after
ploughing, whereas the $CO_2$ uptake of the site considerably increased when compared to post-
restoration years. In detail, we observed large losses of $CO_2$ and $N_2O$ during the year of
restoration. In contrast, the grassland acted as a carbon sink under usual management, i.e. the
time periods (2010-2011 and 2013-2014). Enhanced emissions/emission peaks of $N_2O$ (defined
as exceeding background emissions $0.21 \pm 0.55$ nmol $m^{-2}$ $s^{-1}$ (SE = 0.02) for at least two
sequential days and the seven-day moving average exceeding background emissions) were
observed for almost seven continuous months after restoration as well as following organic
fertilizer applications during all years. Net ecosystem exchange of $CO_2$ ($NEE_{CO2}$) showed a
common pattern of increased uptake of $CO_2$ in spring and reduced uptake in late fall. $NEE_{CO2}$
dropped to zero and became positive after each harvest event. Methane ($CH_4$) exchange
fluctuated around zero during all years. Overall, $CH_4$ exchange was of negligible importance
for both, the GHG budget as well as for the carbon budget of the site.
Our results stress the inclusion of grassland restoration events when providing cumulative sums
of C sequestration potentials and/or global warming potentials (GWPs). Consequently, this
study further highlights the need for continuous long-term GHG exchange observations as well
as the implementation of our findings into biogeochemical process models to track potential
GHG mitigation objectives as well as to predict future GHG emission scenarios reliably.








# 1 Introduction

Grassland ecosystems are commonly known for their provisioning of forage, either directly via grazing of animals on site, or indirectly by regular biomass harvest and preparation of silage or hay. Simultaneously, grasslands have further been acknowledged for their greenhouse gas (GHG) mitigation and soil carbon sequestration potential (Lal, 2004; Smith et al., 2008). However, greenhouse gas emissions from grasslands, particularly $N_2O$ and $CH_4$ have been shown to offset net carbon dioxide equivalent ($CO_2$-eq.) gains (Ammann et al., 2020; Dengel et al., 2011; Hörtnagl et al., 2018; Hörtnagl and Wohlfahrt, 2014; Merbold et al., 2014; Schulze et al., 2009). Still, datasets containing continuous measurements of all three major GHGs ($CO_2$, $CH_4$ and $N_2O$) in grassland ecosystems remain limited (Hörtnagl et al., 2018), include a single GHG only, or focus on specific management activities (Fuchs et al., 2018; Krol et al., 2016). At the same time such datasets are extremely valuable by providing key training datasets for biogeochemical process models (Fuchs et al., 2020a).

Here we investigate the GHG exchange of the three major trace gases ($CO_2$, $CH_4$ and $N_2O$) over five consecutive years in a typical managed grassland on the Swiss plateau. Our study includes the application of traditional GHG chamber measurements and state-of-the-art GHG concentration measurements with a quantum cascade laser absorption spectrometer and a sonic anemometer in an eddy covariance setup (Eugster and Merbold, 2015). Prior to our measurements we hypothesized short-term losses of $CO_2$ and more continuous losses of primarily $N_2O$ following dramatic managements events such as ploughing occurring at irregular time intervals. We further hypothesized an increased carbon uptake strength compared to the pre-ploughing years. Methane emissions were hypothesized to be of minor importance due to the limited time of grazing animals on site (Merbold et al., 2014).

Up to date the majority of greenhouse gas exchange research has focused on $CO_2$, with less focus on the other two important GHGs $N_2O$ and $CH_4$, even though an increased interest in these other gas species has become visible in recent years (Ammann et al., 2020; Ball et al., 1999; Cowan et al., 2016; Krol et al., 2016; Kroon et al., 2007, 2010; Necpálová et al., 2013; Rutledge et al., 2017). The existing exceptions are often referred to as "high-flux" ecosystems, namely wetlands and livestock production system in terms of $CH_4$ (Baldocchi et al., 2012; Felber et al., 2015; Laubach et al., 2016; Teh et al., 2011) and agricultural ecosystems such as bioenergy system with considerable $N_2O$ emissions (Cowan et al., 2016; Fuchs et al., 2018; Krol et al., 2016; Skiba et al., 1996, 2013; Wecking et al., 2020; Zenone et al., 2016; Zona et

al., 2013). Agricultural ecosystems and specifically grazed systems are characterized by GHG
emissions caused through anthropogenic activities. These activities lead to changes in GHG
emission patterns and include harvests, amendments of fertilizer and/or pesticides and less
frequently occurring ploughing, harrowing and re-sowing events. While ploughing has been
shown to lead to considerable short-term emissions of $CO_2$ and $N_2O$ (Buchen et al., 2017;
Cowan et al., 2016; Hörtnagl et al., 2018; MacKenzie et al., 1997; Merbold et al., 2014;
Rutledge et al., 2017; Vellinga et al., 2004), regular harvests have been shown to lead to
increased $CO_2$ uptake (Zeeman et al., 2010) and grazing leads to large $CH_4$ emissions (Dengel
et al., 2011; Felber et al., 2015). Other studies showed contrary results with reduced $N_2O$
emissions following ploughing of a drained grassland when compared to a fallow in Canada
(MacDonald et al., 2011).
Still, the full range of management activities occurring in intensively managed grasslands and
their respective impact on GHG exchange has not been investigated in detail. In a recent
synthesis including grasslands located along an altitudinal gradient in Central Europe, Hörtnagl
et al. (2018) highlighted the most important abiotic drivers of $CO_2$ (light, water availability and
temperature), $CH_4$ (soil water content, temperature and grazing) and $N_2O$ exchange (water
filled pore space and soil temperature). The study by Hörtnagl et al. (2018) further elaborated
the variation in management intensity and related variations in GHG exchange across sites,
stressing the need for more case studies based on continuous GHG observations to improve
existing knowledge and close remaining knowledge gaps. To complete the picture on factors
driving ecosystem GHG exchange, irregular occurring events such as dry spells or
extraordinary wet periods can further lead to enhanced or reduced GHG emissions (Chen et al.,
2016; Hartmann and Niklaus, 2012; Hopkins and Del Prado, 2007; Mudge et al., 2011; Wolf
et al., 2013).
While drought has been shown to reduce $CO_2$ uptake in forests (Ciais et al., 2005) whereas
dry spells did not affect $CO_2$ uptake in grasslands (Wolf et al., 2013), flooding leads primarily
to enhanced $CH_4$ emissions (Knox et al., 2015) and large precipitation events can lead to
plumes of $N_2O$ (Fuchs et al., 2018; Zona et al., 2013) similar to freeze-thaw events (Butterbach-
Bahl et al., 2011; Matzner and Borken, 2008) to name only some examples. Consequently,
understanding both, anthropogenic impacts such as management besides environmental
impacts on ecosystem GHG exchange, are crucially important to suggest appropriate climate
change mitigation as well as adaptation strategies for future land management with ongoing
climate change.

Different measurement techniques to quantify the net GHG exchange in ecosystems are known and the most common approaches are either GHG chamber measurements or the eddy covariance (EC) technique. Static manual chamber measurements have been used for more than a century to quantify $CO_2$ emissions (Lundegardh, 1927) and their application has further been expanded during the last decades to quantify losses of the three major GHGs, $CO_2$, $N_2O$ and $CH_4$ from soils (Imer et al., 2013; Pavelka et al., 2018a; Pumpanen et al., 2004; Rochette et al., 1997). Even though more complex in technology and assumptions made before carrying out measurements, the eddy covariance (EC) technique has become a valuable tool to derive ecosystem integrated $CO_2$ and $H_2O_{vapour}$ exchange across the globe (Baldocchi, 2014; Eugster and Merbold, 2015). The technique has been further extended to continuous measurements of $CH_4$ and $N_2O$ with the development of easy field-deployable fast-response analyzers during the last decade (Brümmer et al., 2017; Felber et al., 2015; Kroon et al., 2007; Nemitz et al., 2018a; Wecking et al., 2020). Each of the two approaches has its strengths and weaknesses and it is beyond the scope of this study to discuss each of them in detail. However, we refer to a set of reference papers highlighting the advantages and disadvantages of each technique separately (chambers: (Ambus et al., 1993; Brümmer et al., 2017; Pavelka et al., 2018a); eddy covariance: (Baldocchi, 2014; Denmead, 2008; Eugster and Merbold, 2015; Nemitz et al., 2018).

The overall objective of this study was to investigate the net GHG exchange ($CO_2$, $CH_4$ and $N_2O$) before and after grassland restoration and thus fill existing knowledge gaps caused by limited amounts of available GHG exchange data from intensively managed grasslands. The specific goals were: (i) to assess pre- and post-ploughing GHG exchange in a permanent grassland in central Switzerland accounting for changes in GHG exchange following frequent management activities; (ii) to compare two different measurement techniques, namely eddy covariance and static greenhouse gas flux chambers to quantify the GHG exchange in a business-as-usual year; and (iii) to provide a five year GHG budget of the site and quantify losses/gains of C and N. Based on our results we provide suggestions for future research approaches to further understand ecosystem GHG exchange, to mitigate GHG emissions and to ensure nutrient retention at the site for sustainable production from permanent grasslands in the future.

## 2 Material and Methods

### 2.1 Study site

The Chamau grassland site (Fluxnet identifier - CH-Cha) is located in the pre-alpine lowlands of Switzerland at an altitude of 400 m a.s.l. (47°12′ 37″N, 8°24′38″E) and characterized by intensive management (Zeeman et al., 2010). The site is divided into two parcels (Parcel A and B) with occasionally slightly different management regimes [see also *Fuchs et al., 2018*]. Mean annual temperature (MAT) is 9.1 °C, and mean annual precipitation (MAP) is 1151 mm. The soil type is a Cambisol with a pH ranging between 5 and 6, a bulk density between 0.9 and 1.3 kg m$^{-3}$ and a carbon stock of 55.5–69.4 t C ha$^{-1}$ in the upper 20 cm of the soil. The common species composition consists of Italian ryegrass (*Lolium multiflorum*) and white clover (*Trifolium repens L.*). For more details of the site we refer to Zeeman et al., (2010).

CH-Cha is intensively managed, with activities being either recurrent – referred to as usual/regular - or sporadic. Usual management refers to regular mowing and subsequent organic fertilizer application in form of liquid slurry (up to 7 times per year). In addition, the site is occasionally grazed by sheep and cattle for few days in early spring and/or fall (H.-R. Wettstein personal communication, Table S1). Sporadic activities aim at maintaining the typical fodder species composition and comprise reseeding, herbicide and pesticide application or irregular ploughing and harrowing on an approximately decadal timescale (Merbold et al., 2014). By such activity, mice are eradicated and a high-quality sward for fodder production is re-established following weed contamination. Specific information on management activity (timing, type of management, amount of biomass harvested) were reported by the farmers on site (Table S1). Additionally, representative samples of organic fertilizer were collected shortly before fertilizer application events and sent to a central laboratory for nutrient content analysis (Labor fuer Boden- und Umweltanalytik, Eric Schweizer AG, Thun, Switzerland). Harvest estimates were compared to estimates based on destructive sampling of randomly chosen plots (n = 10) in the years 2010, 2011, 2013 and 2014. The amount of harvested biomass in the year 2012 was based on a calibration of the values presented by the farmer in comparison to the on-site destructive harvests in previous and following years (Table S1).

### 2.2 Eddy covariance flux measurements

*2.2.1 Eddy covariance setup*

The specific site characteristics with two prevailing wind directions (North-northwest and South-south east) allows continuous observations of both management parcels. It is

noteworthy, that the separation of the two parcels is done exactly at the location of the tower.
See Zeeman et al. (2010) and Fuchs et al. (2018) for further details. The eddy covariance setup
consisted of a three-dimensional sonic anemometer (2.4 m height, Solent R3, Gill Instruments,
Lymington, UK), an open-path infrared gas analyzer (IRGA, LI-7500A, LiCor Biosciences,
Lincoln, NE, USA) to measure the concentrations of $CO_2$ and $H_2O_{vapour}$ and a recently
developed continuous-wave quantum cascade laser absorption spectrometer (mini-QCLAS -
$CH_4$, $N_2O$, $H_2O$ configuration, Aerodyne Research Inc., Billerica, MA, USA) to measure the
concentrations of $CH_4$, $N_2O$, and $H_2O_{vapour}$. 3D wind components (u, v, w), $CO_2$ and $H_2O_{vapour}$
concentration data from the IRGA were collected at a 20 Hz time interval, whereas
concentrations of $CH_4$ and $N_2O$ were collected at a 10 Hz rate from the QCLAS. The QCLAS
provided the dry mole fraction for both trace gases ($CH_4$ and $N_2O$), and data were transferred
to the data acquisition system (MOXA embedded Linux computer, Moxa, Brea, CA, USA) via
an RS-232 serial data link and merged with the sonic anemometer and IRGA data streams in
near-real time (Eugster and Plüss, 2010). Important to note is that the QCLAS was stored in a
temperature-controlled box (temperature variation during the course of a single day was
reduced to < 2 K) and located approximately 4 meters away from the EC tower to avoid long
tubing. Total tube length from the inlet near the sonic anemometer to the measurement cell was
6.5 m. The inlet consisted of a coarse sinter filter (common fuel filter used in model cars) and
a fine vortex filter (mesh size 0.3μm and a water trap) installed directly before the QCLAS.
Filters were changed monthly or if the cell pressure in the laser dropped by more than 2 torr.
Flow rate of approximately $15\,l\,min^{-1}$ was achieved with a large vacuum pump (BOC Edwards
XDS-35i, USA and TriScoll 600, Varian Inc., USA – the latter was used during maintenance
of the Edwards pump). The pumps were maintained annually and replaced twice due to
malfunction during the observation period. The infrared gas analyzer was calibrated to known
concentrations of $CO_2$ and $H_2O$ each year. The QCLAS did not need calibration due to its
operating principles, and an internal reference cell (mini-QCL manual, Aerodyne Research
Inc., Billerica, MA, USA) eased finding the absorption spectra after each restart of the analyzer.

*2.2.2 Eddy covariance flux processing, post-processing and quality control*
Raw fluxes of $CO_2$, $CH_4$, $N_2O$ ($F_{GHG}$, $\mu mol\ m^{-2}\ s^{-1}$) were calculated as the covariance between
turbulent fluctuations of the vertical wind speed and the trace gas species mixing ratio,
respectively (Baldocchi, 2003; Eugster and Merbold, 2015). Open-path infrared gas analyzer
(IRGA) $CO_2$ measurements were corrected for water vapor transfer effects (Webb et al., 1980).
A 2-dimensional coordinate rotation was performed to align the coordinate system with the
mean wind streamlines so that the vertical wind vector $\acute{w} = 0$. Turbulent departures were
calculated by Reynolds (block) averaging of 30 min data blocks. Frequency response
corrections were applied to raw fluxes, accounting for high-pass and low-pass filtering for the
$CO_2$ signal based on the open-path IRGA as well as for the closed-path $CH_4$ and $N_2O$ data
(Fratini et al., 2014). All fluxes were calculated using the software *EddyPro* (version 6.0, LiCor
Biosciences, Lincoln, NE, USA) (Fratini and Mauder, 2014).
The quality of half-hourly raw time series was assessed during flux calculations following
(Vickers and Mahrt, 1997). Raw data were rejected if (a) spikes accounted for more than 1 %
of the time series, (b) more than 10 % of available data points were significantly different from
the overall trend in the 30 min time period, (c) raw data values were outside a plausible range
($\pm$ 50 $\mu$mol m$^{-2}$ s$^{-1}$ for $CO_2$, $\pm$ 300 nmol m$^{-2}$ s$^{-1}$ for $N_2O$ and $\pm$ 1 $\mu$mol m$^{-2}$ s$^{-1}$ for $CH_4$) and (d)
window dirtiness of the IRGA sensor exceeded 80 %. Only raw data that passed all quality
tests were used for flux calculations.
Half-hourly flux data were rejected if (e) fluxes were outside a physically plausible range (ie.
+/- 50 $\mu$mol m$^{-2}$ s$^{-1}$ for $CO_2$) (f) the steady state test exceeded 30 % and (g) the developed
turbulent conditions test exceeded 30 % (Foken et al., 2006). Between 1$^{st}$ January 2010 and
31$^{st}$ December 2014 64572 (88% of all possible data) 30-min flux values were calculated for
$CO_2$, of which 42865 (57.8%) passed all quality tests and were used for analyses in the present
study (Table 1). The amount of available flux values for $N_2O$ and $CH_4$ were less, since we were
only capable to continuously measure both gases from 2012 onwards (Table 1). Flux values in
this manuscript are given as number of moles of matter/mass per ground surface area and unit
time. Negative fluxes represent a flux of a specific gas species from the atmosphere into the
ecosystem, whereas positive fluxes represent a net loss from the system.

**2.3 Static greenhouse gas flux chambers**
*2.3.1 Manual static GHG chamber setup*
Static manual opaque GHG chambers were installed within the footprint of the site to measure
soil fluxes in 2010 and 2011 (n =16) as well as during summer 2013 (n = 10). The chambers
were made of polyvinyl chloride tubes with a diameter of 0.3 m (Imer et al., 2013). The average
headspace height was 0.136 m $\pm$ 0.015 m and average insertion depth of the collars into the
soil was 0.08 m $\pm$ 0.05 m. During sampling days with vegetation larger than 0.3 m inside the
chamber, collar extensions (0.45 m) were used (2013 only). Chamber lids were equipped with
reflective aluminium foil to minimize heating inside the chamber during the period of actual
measurement. Spacing between the chambers was approximately seven m and an equal number
of chambers were installed in each parcel. For further details we refer to Imer et al. (2013).
Chamber measurements were carried out on a weekly basis during the growing season in all
three years (2010, 2011 and 2013), and at least once a month during the winter season in 2010
and 2011. More frequent measurements of $N_2O$ emissions (every day) were performed
following fertilization events in 2013 for seven consecutive days after each event. Besides this,
an intensive measurement campaign lasting 48 hours (two-hour measurement interval) was
carried out in September 2010.
*2.3.2 GHG concentrations measurements*
During each chamber closure four gas samples were taken, one immediately after closure and
then in approximately ten-minute time increments. With this approach, we guaranteed that the
chambers were closed no longer than 40 minutes to avoid potential saturation effects. Syringes
(60 ml volume) were inserted into the chambers lid septa to take the gas samples. The collected
air sample was injected into pre-evacuated 12 ml vials (Labco Limited, Buckinghamshire, UK)
in the next step. Prior to the second, third and fourth sampling of each chamber, the air in
chamber headspace was circulated with the syringe volume of air from the chamber headspace
to minimize effects of built-up concentration gradients inside the chamber.
Gas samples were analyzed for their respective $CO_2$, $CH_4$ and $N_2O$ concentrations in the lab as
soon as possible after sample collection and not stored for more than a few days. Gas sample
analysis was performed with a gas chromatograph (Agilent 6890 equipped with a flame
ionization detector, a methanizer - Agilent Technologies Inc., Santa Clara, USA - and an
electron capture detector – SRI Instruments Europe GmbH, 53604 Bad Honnef, Germany) as
described by Hartmann and Niklaus (2012).
*2.3.3 GHG chamber flux calculations and quality control*
GHG fluxes were calculated based on the rate of gas concentration change inside the chamber
headspace. Data processing, which included flux calculation and quality checks, was carried
out with the statistical software R (R Development Core Team, 2010). Thereby the rate of
change was calculated by the slope of the linear regression of gas concentration over time. Flux
calculation was based on the common equation containing GHG concentration (c in nmol $mol^{-1}$
$^1$ for $CH_4$ and $N_2O$), time (t in seconds), atmospheric pressure (p in Pa), the headspace volume
(V in $m^{-3}$), the universal gas constant (R = 8.3145 $m^{-3}$ Pa $K^{-1}$ $mol^{-1}$), ambient air temperature
(Ta in K) and the surface area enclosed by the chamber (A in $m^{-2}$) (equation 1 in Imer et al.
301 (2013)).

Flux quality criteria were based on the fit of the linear regression. If the correlation coefficient
of the linear regression ($r^2$) was < 0.8 the actual flux value was rejected from the subsequent
data analysis. Furthermore, if the slope between the 1st and 2nd GHG concentration
measurement deviated considerably from the following concentrations we omitted the first
value and calculated the flux based on three instead of four samples. Mean chamber GHG
fluxes were then calculated as the arithmetic mean of all available individual chamber fluxes
for each date. A total of 60 GHG flux calculations ($CH_4$ and $N_2O$) were available for the years
2010 and 2011. Another 52 $N_2O$ flux values were available for the five-month peak-growing
season in 2013.

*2.4 Gapfilling and annual sums of $CO_2$, $CH_4$, and $N_2O$*
To date a common strategy to fill gaps in EC data of $CH_4$ and $N_2O$ has not been agreed on. The
commonly used methods are simple linear approaches (Mishurov and Kiely, 2011) or the
application of more sophisticated tools such as artificial neural networks (Dengel et al., 2011).
The difficulty of finding an adequate gap-filling strategy results from the fact that emission
pulses of either $N_2O$ or $CH_4$ remain challenging to predict. Similarly, different measurement
approaches – i.e. low temporal resolution manual GHG chambers compared to high temporal
resolution eddy covariance measurements - need different gap-filling approaches (Mishurov
and Kiely, 2011; Nemitz et al., 2018). In order to keep the gap-filling methods as simple and
reliable as possible, we used a running median (30 and 60 days for eddy covariance based and
chamber $N_2O$ fluxes, respectively). A similar approach was recently chosen by Hörtnagl et al.
(2018) due to its sensitivity to peaks in the $N_2O$ exchange data. The approach was particularly
chosen as it minimizes the bias occuring from linear gap filling or simply using an overall
average value. While the gapfilling approach may be of less importance for EC flux
measurements with its high temporal data availability, it is the more important for less
frequently available GHG fluxes derived via manual chambers. Given the occurrence of
sporadic $N_2O$ peaks which occur mostly in relation to management activities and last for few
hours/days only as well as the labour needed to carry out GHG chambers measurements,
researchers commonly aim at having weekly or biweekly flux data (i.e. Imer et al. 2013). The
respective sampling design is commonly designed to capture potential $N_2O$ flux peaks as well
as some background values (Mishurov and Kiely, 2011). If one then uses either a linear
interpolation or an overall average value, one can derive a budget which is than a likely
overestimation of the annual flux budget caused by the few flux peaks observed in such
managed systems. The same bias is likely to occur if just flux averages are used since few very

high emission peaks will affect such an average. Thus, and in order to simulate $N_2O$ emission peaks more reliably, we have chosen the approach as taken by Hörtnagl et al. (2018).

In contrast to $CH_4$ and $N_2O$ various well-established approaches to fill $CO_2$ flux data exist (Moffat et al., 2007). Here, we filled gaps in $CO_2$ exchange data following the marginal distribution sampling method (Reichstein et al., 2005) which was implemented in the R package REddyProc (https://r-forge.r-project.org/projects/reddyproc/).

Calculation of the global warming potential (GWP) given in $CO_2$-equivalents followed the recommendations given in the 5[th] Assessment Report of the Intergovernmental Panel on Climate Change (IPCC), with $CH_4$ having a 28 and $N_2O$ a 265 times greater GWP than $CO_2$ on a per mass basis over a time horizon of 100 years (Stocker et al., 2013).

*2.5 Meteorological and phenological data*

Flux measurements were accompanied by standard meteorological measurements. These included observations of soil temperature (depths of 0.01, 0.02, 0.05, 0.10, and 0.15 m, TL107 sensors, Markasub AG, Olten, Switzerland), soil moisture (depths of 0.02 and 0.15 m, ML2x sensors, Delta-T Devices Ltd., Cambridge, UK) and air temperature (2 m height, Hydroclip S3 sensor, Rotronic AG, Switzerland). Furthermore, we measured the radiation balance including short-wave incoming and outgoing radiation, long-wave incoming and outgoing radiation (CNR1 sensor with ventilated Markasub housing, Kipp and Zonen, Delft, the Netherlands) as well as photosynthetically active radiation at 2 m height (PARlite sensor, Kipp and Zonen, Delft, the Netherlands). All data were stored as 30 min averages on a datalogger in a climate-controlled box on site (CR10X, Campbell Scienctific, Logan, UT, USA).

## 3 Results

*3.1 General site conditions*

The Chamau study site (CH-Cha) experienced meteorological conditions typical for the site during the five-year observation period. Summer precipitation commonly exceeded winter precipitation (Figure 1a). A spring drought was recorded from March till May 2011 (Wolf et al., 2013), leading to considerably lower soil water content than in previous and following years (Figure 1a). Average daily air temperatures rose up to 26.7 °C (27th July 2013) during summer and average daily temperature in winter dropped as low as -12.7 °C (6th February 2012, Figure 1b) with soil temperature following in a dampened pattern (Figure 1b). Average daily photosynthetic photon flux density did not differ considerably over the five-year observation period (Figure 1c). The site rarely experienced snow cover during winter (Figure 1b).

The complexity in management activities becomes apparent when comparing business as usual years (e.g. 2011) with the restoration year (2012, Figure 2a and b), highlighting the importance of grassland restoration to maintain productivity yields. Prior to 2012 an obvious decline in productivity with larger C and N inputs was found compared to the outputs in the years after restoration (2013 and 2014, Figure 2a and b).

*3.2 EC $N_2O$ fluxes vs. chamber derived $N_2O$ fluxes*

In 2013, we had the chance of comparing $N_2O$ fluxes measured with two considerably different GHG measurement techniques, namely eddy covariance and static chambers. The chambers (n=10) were installed within the EC footprint. Our results reveal a similar temporal pattern, with increased $N_2O$ losses being captured by both methodologies following fertilizer application. However, we could not identify a consistent bias of either technique (Figure 3a). Direct comparison of both measurements revealed a reasonable correlation (slope m = 0.61, $r^2$ = 0.4) and larger variation between both techniques with increasing flux values (Figure 3b).

*3.3 Temporal variation of GHG exchange*

Fluxes of $CO_2$ and $N_2O$ showed considerable variation between and within years. This variation primarily occurs due to management activities and seasonal changes in meteorological variables (Figures 1 and 4). In contrast, methane fluxes did not show a distinct seasonal pattern.

*CO₂ exchange*

In pre-ploughing years (2010 and 2011), the Chamau site showed 60 % lower $CO_2$ uptake compared to the post-ploughing years (2013 and 2014, Table 2). All four non-ploughing years revealed largest $CO_2$ uptake rates in late spring (daily averaged peak uptake rates were >10 μmol $CO_2$ m$^{-2}$ s$^{-1}$, March and April, Figure 4a). Besides the seasonal effects a clear impact of harvest events could be identified, with abrupt changes from net uptake of $CO_2$ to either reduced uptake or net loss of $CO_2$ (light blue arrows indicate harvest event, Figure 4a). A similar but less pronounced effect was found following grazing periods (light and dark brown arrow, Figure 4a). A complete switch from net uptake to net $CO_2$ release was observed during the first three months of 2012, after ploughing and during re-cultivation of the grassland. In this specific year, the site only experienced snow cover for few days (Figure 1c) and temperatures below 5 ºC occurred more regularly than in all other years (Figure 1 b). Seasonal $CO_2$ exchange was characterized by net release of $CO_2$ in winter (DJF), highest $CO_2$ uptake rates were observed in spring (MAM), constant uptake rates during summer (JJA) which however were lower than those measured in spring, and very low net release of $CO_2$ in fall (Table 3). Average winter $CO_2$ exchange for the five-year observation period (gap-filled 30 min data) was 0.28 ± 5.68 μmol $CO_2$ m$^{-2}$ s$^{-1}$ (SE = 0.04, Table 3). The restoration year 2012 showed a slightly different pattern with relatively large $CO_2$ release in winter and spring and considerably lower uptake rates in summer. The years before the restoration (2010 and 2011) were characterized by smaller net uptake rates during spring and summer when compared to the post-ploughing years (2013 and 2014). Additionally, winter fluxes in 2010 and 2011 were positive (net release of $CO_2$), while winter fluxes in the years 2013 and 2014 were showing a small but consistent net uptake of $CO_2$ (Figure 4a, Table 3).

*CH₄ exchange*

The individual static chamber measurements (2011&2011) were often below the detection limit and fluctuated around zero similar to the eddy covariance measurements (Figure 4b). Any methane peaks expected due to freezing and thawing in late winter and early spring were not observed. Also, commonly reported net emissions of methane during grazing of animals were not seen (Figure 4b). Seasonal differences of methane exchange did not show a clear pattern (Table 3). A comparison of methane fluxes obtained by both, static GHG chambers and EC measurements as done for $N_2O$ (see next paragraph) could not be performed due to a malfunction of the respective detector in the gas chromatograph.

*N$_2$O exchange*

N$_2$O exchange was low during the majority of the days over the five-year observation period, fluctuating around zero (Figure 4c). However, clear peaks in N$_2$O emissions were observed following fertilization events or periods with high rainfall after a dry period in summer (i.e. summer 2013 and 2014, Figures 3a and 4c). While event driven N$_2$O emissions were commonly on the order of 4 to 8 nmol N$_2$O m$^{-2}$ s$^{-1}$ (Figure 4c), N$_2$O emissions following ploughing and subsequent re-sowing of the grassland in 2012 lead to up to three times as high N$_2$O emissions (Figure 4c, year 2012, see also Merbold et al. (2014)). Similar to methane, enhanced N$_2$O emissions in late winter or early spring as reported by other studies could not be identified (Figure 4c).

Background N$_2$O fluxes were estimated by analysing all high temporal resolution flux data but excluding the restoration year 2012 and all values one week after a management event. Daily average background fluxes were $0.21 \pm 0.55$ nmol m$^{-2}$ s$^{-1}$ (SE = 0.02). Differences in N$_2$O exchange over the course of individual years became obvious when splitting the dataset into the four seasons (winter – DJF, spring – MAM, summer – JJA and fall – SON). In contrast to CO$_2$ exchange that showed large net uptake rates in spring, N$_2$O emissions were largest during summer (JJA) and lowest in winter (DJF). As highlighted for the other gases, the year of grassland restoration showed a completely different picture (Table 3).

*3.4 Annual sums and Global Warming Potential (GWP) of CO$_2$, CH$_4$ and N$_2$O*

Annual sums showed a net uptake of CO$_2$ during the two pre-ploughing years (-695 g CO$_2$ m$^{-2}$ yr$^{-1}$ and -978 g CO$_2$ m$^{-2}$ yr$^{-1}$ in 2010 and 2011 respectively). Up to three times of this net uptake was reached in 2013 and 2014, the two post-ploughing years (-2046 g CO$_2$ m$^{-2}$ yr$^{-1}$ and -2751 g CO$_2$ m$^{-2}$ yr$^{-1}$, Table 2). In contrast, the ploughing year 2011 was characterized by a net release of CO$_2$ (1447 g CO$_2$ m$^{-2}$ yr$^{-1}$).

Methane budgets for the years 2010 and 2011 were not be calculated as many of the available measurements were below the limit of detection. For the years 2012 – 2014, the annual methane budget showed a minor release of 26.8 – 55.2 g CH$_4$ m$^{-2}$ yr$^{-1}$.

The Chamau site was characterized by a net release of nitrous oxide over the five-year study period. While annual average N$_2$O emissions ranging between 0.34 and 1.17 g N$_2$O m$^{-2}$ yr$^{-1}$ in the non-ploughing years, the site emitted 4.36 g N$_2$O m$^{-2}$ yr$^{-1}$ in 2012.

The global warming potential (GWP), expressed as the yearly cumulative sum of all gases after their conversion to CO$_2$-equivalents, was negative during all years (between -387 and -2577 CO$_2$-eq. m$^{-2}$) except for the ploughing year 2012 (+2629 CO$_2$-eq. m$^{-2}$).

Overall, $CO_2$ exchange contributed more than 90% to the total GHG balance in 2011, 2013 and
2014. Clearly, $CH_4$ exchange was of minimal importance for the GHG budget (Table 2). In
2010, the contribution of $CO_2$ to the site's GHG budget was almost 70%, and $N_2O$ contributed
about 30%. Only in 2012, the year of restoration, $CO_2$ and $N_2O$ exchange contributed almost
equally to the site's overall GHG budget (55.1% and 43.9%, respectively).

*3.5. Carbon gains/losses of the Chamau site between 2010 and 2014*
The Chamau site assimilated on average -441 ± 260 g $CO_2$-C $m^{-2}$ $yr^{-1}$ (4410 kg C $ha^{-1}$ $yr^{-1}$)
during the "business as usual" years (2010 and 2011 as well as 2013 and 2014). During the
restoration year the site lost 395 g $CO_2$-C $m^{-2}$ (3950 kg C $ha^{-1}$) (Table 2). Carbon losses (and/or
gains) from methane were < 1 g $CH_4$-C $m^{-2}$ during all five years.
Carbon was gained in both parcels during the pre-ploughing years (Table 4). Considerable net
losses of carbon were calculated for the ploughing year. In contrast, the post-ploughing years
were again recognized as years with large net gains in carbon. Over the observation period of
5 years, the Chamau grassland gained approximately 4 t C $ha^{-1}$, excluding losses via leaching
and deposition of C in form of dust.

















## 4 Discussion

The five-year measurement period is representative for other similarly managed grassland ecosystems in Switzerland. Climate conditions were similar to the long-term average as described in Wolf et al. (2013). Management activities, such as harvests and subsequent fertilizer applications, were driven by overall weather conditions, (i.e. 2013 late spring, Figure 2a and b).

*4.1 Technical and methodological aspects of the study*

Different techniques are currently applied to measure GHG fluxes from a variety of ecosystems (Denmead, 2008), each having its advantages and disadvantages or being chosen for a specific purpose or reason. A common approach to study individual processes or time periods contributing to specific greenhouse gas emissions is to measure with GHG chambers on the plot scale (Pavelka et al., 2018). Chamber methods have been widely used to derive annual GHG and nutrient budgets (Barton et al., 2015; Butterbach-Bahl et al., 2013). Critical assessments of the suitability and associated uncertainty in chamber derived GHG budgets in relation to sampling frequency have been published by Barton et al. (2013). Existing studies have not only compared the two measurement techniques employed in this study (manual chambers and eddy covariance) in grasslands before, but also estimated annual emissions based on differing methodologies (Flechard et al., 2007; Jones et al., 2017). Additional confidence in our approach was obtained from the $N_2O$ emissions during the summer period 2013, where both measurement techniques ran in parallel (Figure 3a and b). Annual budgets derived by applying similar gap-filling approaches to the individual datasets led to comparable results (Table 2).

We calculated detection limits for the individual GHGs from our manual chambers following (Parkin et al., 2012). Detection limits were $0.34 \pm 0.26$ nmol m$^{-2}$ s$^{-1}$, $0.05 \pm 0.02$ nmol m$^{-2}$ s$^{-1}$, and $0.06 \pm 0.06$ μmol m$^{-2}$ s$^{-1}$ for $CH_4$, $N_2O$ and $CO_2$, respectively. Following this, methane flux measurements frequently were below this limit of detection, hence we did not calculate methane budgets for 2010 and 2011. The flux values measured with the EC technique between 2012 and 2014 compare well to similar measurements made by Felber et al. (2016) in an intensively managed grassland in Western Switzerland. The observed values have been identified to represent the soil methane exchange in EC measured fluxes (Felber et al. 2016). $N_2O$ fluxes in contrast were much better constrained by both methods due to clear $N_2O$ sources (i.e. fertilizer amendments) and better sensitivity of the instruments used by both techniques

for $N_2O$ as compared to $CH_4$. Background $N_2O$ emissions as observed in this study (0.21 ±
0.55 nmol $m^{-2}$ $s^{-1}$ (SE = 0.02)) compare well to estimates suggested by Rafique et al., (2011)
whom suggest an annual background $N_2O$ losses of 1.8 kg $N_2O$-N for a grazed pasture (i.e.
0.20 nmol $m^{-2}$ $s^{-1}$).

*4.2 Annual GHG and C and N gains/losses*
Net carbon losses and gains estimated for the CH-Cha site between 2010 and 2015 were in
general within the range of values estimated by Zeeman et al., (2010) for the years 2006 and
2007. The slightly higher losses observed prior to ploughing may result from reduced
productivity of the sward. This becomes particularly visible when compared to the net
ecosystem exchange (NEE) of $CO_2$ values for the years after restoration. Losses via leaching
have previously been estimated to be of minor importance at this site (Zeeman et al., 2010) and
were therefore not considered in this study. Considerably higher C gains during post-ploughing
years were caused be enhanced plant growth in spring and summer. Restoration is primarily
done to eradicate weeds and rodents, favouring biomass productivity of the fodder grass
composition. Other grasslands in Central Europe, i.e. sites in Austria, France and Germany,
showed similar values for net ecosystem exchange (Hörtnagl et al., 2018). Still, total C budgets
as presented here are subject to considerable uncertainty which is strongly depending on
assumptions made for gap-filling etc. (Foken et al., 2004). Nevertheless, the values reported
here show the overall trend on C uptake/release of the site and clearly exceed the uncertainty
of ± 50 g C per year for eddy covariance studies as suggested by Baldocchi (2003).
Methane was of negligible importance for the C budget of this site. We did not observe distinct
peaks in $CH_4$ emissions in relation to grazing which is primarily due to the low grazing pressure
at CH-Cha. Studies carried out on pastures in Scotland, Mongolia, France and Western
Switzerland have shown that grazing can largely contribute to ecosystem-scale methane fluxes,
in particular if ruminants such as cattle are populating the EC footprint (Dengel et al., 2011;
Felber et al., 2015; Schönbach et al., 2012). If we included an approximation of methane
emissions of cattle which we may have missed in the EC flux measurements, we would have
to add 0.407 g $CH_4$-C $m^{-2}$ $y^{-1}$ to the current value of 1.48 g $CH_4$-C $m^{-2}$ in 2014 (Table 2). This
value is based on the average methane emissions of 404 g $CH_4$ $head^{-1}$ $d^{-1}$ stated in Felber et al.
(2016) and linking this to the average stocking density (4.04 head ha-1) on the Chamua site
and the stocking duration (30 days in 2014). Still, the GHG budget as well as the C budget of
the site would not be altered.
The nitrous oxide budget reported for the years without ploughing in this study coincides with
values reported for other grasslands in Europe, ranging from moist to dry climates and lower
to higher elevations in Austria and Switzerland (Cowan et al., 2016; Hörtnagl et al., 2018; Imer
et al., 2013; Skiba et al., 2013).
Nitrogen inputs and losses via $N_2O$ varied largely between the years before and after ploughing.
While the site was characterized by large N amendments prior to ploughing and with reduced
harvest, the picture was completely the opposite during the years after ploughing, with
considerably less N inputs compared to the nitrogen removed from the field via harvests.
Farmers aim every year at having a balanced N budget (fertilizer inputs = nutrients removed
from the field). Pasture degradation is the main motivation for enhanced fertilizer inputs in
order to stabilize forage productivity. Similarly, regular restoration of permanent pastures is
absolutely necessary (Cowan et al., 2016). So far, we identified only one study that investigated
the net effects on the overall GHG exchange following grassland restoration (Drewer et al.,
583 2017).


**5 Conclusion**
This study in combination with an overview of available datasets on grassland restoration and
their consequences on GHG budgets highlights the overall need of additional observational
data. While restoration changed the previous C sink to a C source at the Chamau site, the wider
implication in terms of the GWP of the site when including other GHGs have long-term
consequences (i.e. in mitigation assessments). Furthermore, this study showed the large
variations in N inputs and N outputs from this grassland and the difficulty farmers face when
aiming for balanced N budgets in the field. Still, the current study focused on GHGs only and
can thus not constrain the N budget but assess the losses of N via $N_2O$. Losses in form of $NH_3$,
$N_2$ and $NO_x$ will have to be quantified to fully assess N budgets besides the overall fact that
GHG data following grassland restoration remain largely limited to investigate long-term
consequences.
Fortunately, these are likely to become available in the near future by the establishment of
environmental research infrastructures (i.e. ICOS in Europe, NEON in the USA or TERN in
Australia) that aim at standardized, high quality and high temporal resolution trace gas
observation of major ecosystems, including permanent grasslands. With these additional data,
another major constraint of producing defensible GHG and nutrient budgets, namely gap-filling
procedures, will likely be overcome. New and existing data can be used to derive reliable

| 603 | functional relations and artificial neural networks (ANNs) at field to ecosystem scale that are |
| 604 | capable of reproducing in-situ measured data. Once this step is achieved, both the available |
| 605 | data as well the functional relations can be used to improve, to train and to validate existing |
| 606 | biogeochemical process models (Fuchs et al., 2020). Subsequently, reliable projections on both |
| 607 | nutrient and GHG budgets at the ecosystem scale that are driven by anthropogenic management |
| 608 | as well as climatic variability become reality. |

| 609 | The study stresses the necessity of including management activities occurring at low frequency |
| 610 | such as ploughing in GHG and nutrient budget estimates. Only then, the effect of potential |
| 611 | best-bet climate change mitigation options can be thoroughly quantified. The next steps in |
| 612 | GHG observations from grassland must not only focus on observing business as usual |
| 613 | activities, but also aim at testing the just mentioned best-bet mitigation options jointly in the |
| 614 | field while simultaneously in combination with existing biogeochemical process models. |

## 6 Tables and Figures

Table 1: Data availability of GHG fluxes measured over the five-year observation period. Values are given as all data possible, raw processed values and high quality (HQ) data, which were then used in the analysis. High quality data are data with a quality flag "0" and "1" from the Eddypro output only. Grey shaded areas represent time period where both methods (EC and static chambers) were used simultaneously to estimate $FN_2O$. Static chamber flux data are highlighted in *italic* font.

**Table 2:** Table 2: Annual average $CO_2$, $CH_4$ and $N_2O$ fluxes and annual sums for the three GHGs as well as carbon and nitrogen gain/losses per gas species. GWP were calculated for a 100-year time horizon and based on the most recent numbers provided by IPCC (Stocker et al., 2013). Annual budgets were derived from either gap-filled manual chamber (MC) or eddy covariance (EC) measurements. n.c. stands for not calculated. Sign convention: positive values denote export/release, negative values import/uptake.

**Table 3:** Average GHG flux rates per season: winter (DJF), spring (MAM), summer (JJA) and fall (SON). Values are based on gap-filled data to avoid bias from missing nighttime data (predominantly relevant for $CO_2$). Data are only presented when continuous measurements (eddy covariance data) were available. Sign convention: positive values denote export/release, negative values import/uptake.

**Table 4:** Table 4: Carbon and nitrogen gains/losses through fertilization, harvest and GHGs for the Chamau (CH-Cha) site in 2010- 2014. Values are given in kg ha$^{-1}$. Gains are indicated

with "-" and losses/exports are indicated with "+". While management information was available for both parcels (A and B), flux measurements are an integrate of both parcels. n.c. = not calculated

**Table 5:** Existing studies investigating the GHG exchange over pastures following ploughing. Results presented show the flux magnitude following ploughing and are rounded values of the individual presented in the papers. Values were converted to similar units (mg $CO_2$-C $m^{-2}$ $h^{-1}$, µg $CH_4$-C $m^{-2}$ $h^{-1}$ and µg $N_2O$-N $m^{-2}$ $h^{-1}$). Based on Web of Knowledge search July 15th 2017 with the search terms "grassland", "pasture", "greenhouse gas", "ploughing" and/or "tillage". Only two studies representing conversion from pasture to cropland or other systems were included in this table.

**Table S1:** Detailed management information for the two parcels under investigation at the Chamau research station. Data are based on fieldbooks provided by the farm personnel as well as in-situ measurements. Organic fertilizer samples were sent to a central laboratory for nutrient content analysis (Labor fuer Boden- und Umweltanalytik, Eric Schweizer AG, Thun, Switzerland). Destructive harvests (n = 10) of biomass were carried out in the years 2010, 2011, 2013 and 2014. Harvest estimates are based on values derived from the in-situ measurements and data provided by the farm personnel. Detailed information on the grazing regime was furthermore provided by the farm personnel in hand-written form (not shown).

**Figure 1:** Weather conditions during the years 2010 – 2014. Weather data were measured with our meteorological sensors installed on site. (a) Daily sum of precipitation (mm) and soil water content (SWC, blue line, $m^3$ $m^{-3}$) measured at 5 cm soil depth; (b) daily averaged air temperature (°C), daily averaged soil temperature (grey line, °C) and days with snow cover (horizontal bars); (c) daily averaged photosynthetic photon flux density (PPFD, µmol $m^{-2}$ $s^{-1}$). Days with snow cover were identified with albedo calculations. Days with albedo > 0.45 were identified as days with either snow or hoarfrost cover.

**Figure 2:** Management activities for both parcels (A and B in panels (a) and (b), respectively) on the CH-Cha site. Overall management varied particularly in 2010 between both parcels, whereas similar management took place between 2011 and 2014. Arrow direction indicates whether carbon (C in kg $ha^{-1}$) and/or nitrogen (N in kg $ha^{-1}$) were amended to, or exported from the site ("$F_o$" and "$F_{o*}$"- organic fertilizers, slurry/manure (red); "$F_m$" - mineral fertilizer (light orange); "H" - harvest (light blue); "$G_s$" and "$G_c$" - grazing with sheep/cows (light/dark brown). Other colored arrows visualize any other management activities such as pesticide application ("$P_h$"- herbicide (light pink); "$P_m$"- molluscicide (dark pink); "T"- tillage (black), "R"- rolling (light grey) and "S"- sowing (dark grey) which occurred predominantly in 2010 (parcel B) and 2012 (parcels A and B). Carbon imports and exports are indicated by black and grey bars. Thereby black indicated the start of the specific management activities and grey the duration (e.g. during grazing, "$G_s$"). Green colors indicate nitrogen amendments or losses, with dark green visualizing the start of the activity and light green colors indicating the duration. Sign convention: positive values denote export/release, negative values import/uptake.

**Figure 3:** (a) Temporal dynamics of $N_2O$ fluxes measured with the eddy covariance (white circles) and manual greenhouse gas chambers (black circles measured in 2013) – grey lines indicate standard deviation. Arrows indicate management events ("H" = harvest, "$F_o$" = organic fertilizer application (slurry), "Ph" = pesticide (herbicide) application). (b) 1:1 comparison between chamber based and eddy covariance based $N_2O$ fluxes in 2013. The

dashed line represents the 1:1 line. ($y = mx + c$, $r^2 = 0.4$, $m = 0.61$, $c = 0.17$, $p < 0.0001$). Sign
convention: positive values denote export/release, negative values import/uptake.
**Figure 4:** Temporal dynamics of gap-filled (except methane in 2010/2011) daily averaged
greenhouse gas (GHG) fluxes (white circles): a) ($CO_2$ exchange in $\mu$mol m$^{-2}$ s$^{-1}$; (b) $CH_4$
exchange in nmol m$^{-2}$ s$^{-1}$ and (c) $N_2O$ exchange in nmol m$^{-2}$ s$^{-1}$. Coloured circles indicate
manual chamber measurements. While both GHGs, $CH_4$ and $N_2O$ were measured in 2010 and
2011 (blue cirlces), $N_2O$ only was measured in 2013 (light blue circles). The grey dashed lines
indicate the beginning of a new year. Same color coding as used in Figure 3 a was used to
highlight management activities. Sign convention: positive values denote export/release,
negative values import/uptake. Grey lines behind the circles indicate standard deviation.

## 7 Acknowledgements

Funding for this study is gratefully acknowledged and was provided by the following projects: Models4Pastures (FACCE-JPI project, SNSF funded contract: 40FA40_154245 / 1), GHG-Europe (FP7, EU contract No. 244122), COST-ES0804 ABBA and SNF-R'EQUIP (206021_133763). We are specifically thankful to Hans-Rudolf Wettstein, Ivo Widmer and Tina Stiefel for providing crucial management data and support in the field. Further, this project could not have been accomplished without the help from the technical team, specifically Peter Plüss, Thomas Baur, Florian Käslin, Philip Meier and Patrick Flütsch. We greatly acknowledge their help during the planning stage, and the endurance during the setup of the new QCLAS system as well as regular trouble shooting of the Swissfluxnet Chamau (CH-Cha) research site.

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

**Table 1:** Data availability of GHG fluxes measured over the five-year observation period. Values are given as all data possible, raw processed values and high quality (HQ) data, which were then used in the analysis. High quality data are data with a quality flag "0" and "1" from the Eddypro output only. Grey shaded areas represent time period where both methods (EC and static chambers) were used simultaneously to estimate FN2O. Static chamber flux data are highlighted in *italic* font.

| Year | $F_{CO2}$ | | | $F_{CH4}$* | | | $F_{N2O}$* | | |
|---|---|---|---|---|---|---|---|---|---|
| | max data availability | raw fluxes | HQ fluxes (0,1) | max data availability | raw fluxes | HQ fluxes (0,1) | max data availability | raw fluxes | HQ fluxes (0,1) |
| **2010** | | | | | | | | | |
| 30min | 17520 | 16064 | 10171 | *365* | *44* | *44* | *365* | *44* | *44* |
| % | 100 | 91.68 | 58.05 | *100* | *12.05* | *12.05* | *100* | *12.05* | *12.05* |
| **2011** | | | | | | | | | |
| 30min | 17520 | 14873 | 10002 | *365* | *16* | *16* | *365* | *16* | *16* |
| % | 100 | 84.8 | 57.08 | *100* | *4.38* | *4.38* | *100* | *4.38* | *4.38* |
| **2012** | | | | | | | | | |
| 30min | 17568 | 15361 | 10165 | 17568 | 15523 | 10181 | 17568 | 15528 | 12859 |
| % | 100 | 87.43 | 57.85 | 100 | 88.35 | 57.95 | 100 | 88.38 | 73.19 |
| **2013** | | | | | | | | | |
| 30min | 17520 | 14825 | 10409 | 17520 | 17200 | 11310 | 17520 *(365)* | 17200 *(52)* | 11790 *(52)* |
| % | 100 | 84.61 | 59.4 | 100 | 98.16 | 64.55 | 100 *(100)* | 98.16 *(14.24)* | 67.29 *(14.24)* |
| **2014** | | | | | | | | | |
| 30min | 17520 | 15719 | 10064 | 17520 | 17207 | 11166 | 17520 | 17207 | 11986 |
| % | 100 | 89.71 | 57.43 | 100 | 98.2 | 63.72 | 100 | 98.2 | 68.4 |
| **All Years** | | | | | | | | | |
| 30min | **87648** | **76842** | **50811** | **87548 *(730)*** | **49930 *(60)*** | **32657 *(60)*** | **87648 *(1826)*** | **49935 *(112)*** | **36635 *(112)*** |
| % | **100** | **87.67** | **57.97** | **100 *(100)*** | **57.03 *(8.22)*** | **37.30 *(8.22)*** | **100 *(100)*** | **57.03 *(6.13)*** | **41.94 *(6.13)*** |

* data availability in parenthesis are based on static manual chambers (2010 and 2011, approx. biweekly measurements (n = 44 and 16 respectively) as well as during summer 2013 (n = 52))

calculated for a 100-year time horizon and based on the most recent numbers provided by IPCC (Stocker et al., 2013). Annual budgets were derived from either gap-filled manual chamber (MC) or eddy covariance (EC) measurements. n.c. stands for not calculated. **Sign convention: positive values denote export/release, negative values import/uptake.**

| | 2010 (MC) | 2010 (EC) | 2011 (MC) | 2011 (EC) | 2012 (MC) | 2012 (EC) | 2013 (MC) | 2013 (EC) | 2014 (MC) | 2014 (EC) |
|---|---|---|---|---|---|---|---|---|---|---|
| Average CO$_2$ flux μmol m$^{-2}$ s$^{-1}$ | -0.5 | -0.7 | | | | 1.04 | | -1.4 | | -1.98 |
| STDEV Average CO$_2$ flux μmol m$^{-2}$ s$^{-1}$ | 3.11 | 3.63 | | | | 3.02 | | 3.52 | | 3.9 |
| g CO$_2$ m$^{-2}$ | -695.23 | -978.16 | | | | 1447.16 | | -2047.8 | | -2751.66 |
| g CO$_2$-C m$^{-2}$ | -189.6 | -266.77 | | | | 394.68 | | -558.49 | | -750.45 |
| **Global warming potential in g CO$_2$-eq. m$^{-2}$** | **-695.23** | **-978.16** | | | | **1447.16** | | **-2047.8** | | **-2751.66** |
| **% of the total budget** | **69.2** | **91.6** | | | | **55.1** | | **92.3** | | **94** |
| Average CH$_4$ flux nmol m$^{-2}$ s$^{-1}$ | n.c. | n.c. | | | | 1.91 | | 3.67 | | 3.92 |
| STDEV Average CH$_4$ flux nmol m$^{-2}$ s$^{-1}$ | n.c. | n.c. | | | | 11.8 | | 9.77 | | 20.61 |
| g CH$_4$ m$^{-2}$ | n.c. | n.c. | | | | 0.96 | | 1.85 | | 1.97 |
| g CH$_4$-C m$^{-2}$ | n.c. | n.c. | | | | 0.72 | | 1.39 | | 1.48 |
| **Global warming potential in g CO$_2$-eq. m$^{-2}$** | **n.c.** | **n.c.** | | | | **26.88** | | **51.8** | | **55.16** |
| **% of the total budget** | **n.c.** | **n.c.** | | | | **1** | | **2.3** | | **1.9** |
| Average N$_2$O flux nmol m$^{-2}$ s$^{-1}$ | 0.84 | 0.25 | | | | 3.13 | 0.28 | 0.32 | | 0.32 |
| STDEV Average N$_2$O flux nmol m$^{-2}$ s$^{-1}$ | 0.84 | 0.2 | | | | 4.35 | 0.6 | 0.73 | | 0.68 |
| g N$_2$O m$^{-2}$ | 1.17 | 0.34 | | | | 4.36 | 0.39 | 0.45 | | 0.45 |
| g N$_2$O-N m$^{-2}$ | 0.74 | 0.22 | | | | 2.77 | 0.25 | 0.28 | | 0.28 |
| **Global warming potential in g CO$_2$-eq. m$^{-2}$** | **310.05** | **90.1** | | | | **1155.4** | **103.35** | **119.25** | | **119.25** |
| **% of the total budget** | **30.8** | **8.4** | | | | **43.9** | | **5.4** | | **4.1** |
| **Total GWP potential** | **-385.18** | **-888.06** | | | | **2629.44** | **-1892.65** | **-1876.75** | | **-2577.25** |

**Table 3:** Average GHG flux rates per season: winter (DJF), spring (MAM), summer (JJA) and fall (SON). Values are based on gap-filled data to avoid bias from missing nighttime data (predominantly relevant for CO2). Data are only presented when continuous measurements (eddy covariance data) were available. Sign convention: positive values denote export/release, negative

| | $CO_2$ ($\mu$mol m$^{-2}$ s$^{-1}$) | | | | $CH_4$ (nmol m$^{-2}$ s$^{-1}$) | | | | $N_2O$ (nmol m$^{-2}$ s$^{-1}$) | | | |
|---|---|---|---|---|---|---|---|---|---|---|---|---|
| | **DJF** | **MAM** | **JAJ** | **SON** | **DJF** | **MAM** | **JAJ** | **SON** | **DJF** | **MAM** | **JAJ** | **SON** |
| **2010** | 0.56 | -1.75 | -0.79 | 0.01 | | | | | | | | |
| **SD** | 5.39 | 12.07 | 11.34 | 9.31 | | | | | | | | |
| **2011** | 0.48 | -4.29 | 0.39 | 0.66 | | | | | | | | |
| **SD** | 5.47 | 10.54 | 12.52 | 8.97 | | | | | | | | |
| **2012** | 0.98 | 3.64 | -0.33 | -0.13 | 2.2 | 1.38 | 2.76 | 1.32 | 3.1 | 5.61 | 3.06 | 0.73 |
| **SD** | 5.69 | 9.1 | 13.65 | 8.03 | 14.91 | 11.85 | 10 | 9.94 | 4.77 | 5.52 | 3.19 | 0.92 |
| **2013** | -0.2 | -4.49 | -1.3 | 0.13 | 2.18 | 5.3 | 3.79 | 3.4 | 0.12 | 0.19 | 0.73 | 0.26 |
| **SD** | 5.04 | 12.98 | 12.14 | 9.81 | 11.31 | 9.25 | 9.08 | 9.21 | 0.23 | 0.37 | 1.27 | 0.38 |
| **2014** | -0.42 | -5.07 | -2.43 | 0.04 | 6.71 | 5.49 | 0.08 | 3.47 | 0.18 | 0.4 | 0.45 | 0.27 |
| **SD** | 6.56 | 12.93 | 12.98 | 9.45 | 22.93 | 31.37 | 8.5 | 1021 | 0.27 | 0.78 | 0.87 | 0.63 |
| **2010-2014** | 0.28 | -2.39 | -0.89 | 0.14 | 3.69 | 4.06 | 2.21 | 2.73 | 1.14 | 2.07 | 1.42 | 0.42 |
| **SD** | 5.68 | 12.06 | 12.58 | 9.14 | 17.15 | 20.11 | 9.31 | 9.81 | 3.09 | 4.08 | 2.35 | 0.71 |

**Table 4:** Carbon and nitrogen gains/losses through fertilization, harvest and GHGs for the Chamau (CH-Cha) site in 2010- 2014. Values are given in kg ha-1. Gains are indicated with "-" and losses/exports are indicated with "+". While management information was available for both parcels (A and B), flux measurements are an integrate of both parcels. n.c. = not calculated

| | 2010 | | 2011 | | 2012 | | 2013 | | 2014 | | Total 2010 - 2014 | |
|---|---|---|---|---|---|---|---|---|---|---|---|---|
| | Carbon | Nitrogen | Carbon | Nitrogen | Carbon | Nitrogen | Carbon | Nitrogen | Carbon | Nitrogen | Carbon | Nitrogen |
| **Fertilizer (kg ha$^{-1}$) - Parcel A** | -1425.53 | -253.09 | -1222.06 | -253.97 | -2242.51 | -271.12 | -926.81 | -213.19 | -385.04 | -122.08 | -6201.95 | -1113.45 |
| **Fertilizer (kg ha$^{-1}$) - Parcel A** | -1487.1 | -194.3 | -1509.9 | -258.3 | -2229 | -293.2 | -1001.1 | -240 | -996.8 | -183.2 | -7223.9 | -1169 |
| | | | | | | | | | | | | |
| **Harvest (kg ha$^{-1}$) - Parcel B** | 3449.26 | 221.85 | 2570.3 | 165.32 | 1684.88 | 108.37 | 4393.9 | 282.61 | 3527.29 | 226.87 | 15625.63 | 1005.02 |
| **Harvest (kg ha$^{-1}$) - Parcel B** | 2018.6 | 129.8 | 1952.2 | 125.6 | 1481.2 | 95.3 | 4174.8 | 268.5 | 6673.4 | 429.2 | 16300.2 | 1048.4 |
| | | | | | | | | | | | | |
| **Flux (CO2-C kg ha$^{-1}$)** | -1896.6 | | -2667.7 | | 3946.8 | | -5584.9 | | -7504.5 | | -13706.9 | |
| **Flux (CH4-C kg ha$^{-1}$)** | n.c. | | n.c. | | 7.2 | | 13.9 | | 14.8 | | 35.9 | |
| **Flux (N2O-N kg ha$^{-1}$)** | | 7.4 | | 2.2 | | 27.7 | | 2.8 | | 2.8 | | 42.9 |
| | | | | | | | | | | | | |
| **Total - Parcel A** | 127.13 | -23.84 | -1319.46 | -86.45 | 3396.37 | -135.05 | -2103.91 | 72.22 | -4347.45 | 107.59 | -4247.32 | -65.53 |
| **Total - Parcel B** | -1365.1 | -57.1 | -2225.4 | -130.5 | 3206.2 | -170.2 | -2397.3 | 31.3 | -1813.1 | 248.8 | -4594.7 | -77.7 |

**Table 5:** Existing studies investigating the GHG exchange over pastures following ploughing. Results presented show the flux magnitude following ploughing and are rounded values of the individual presented in the papers. Values were converted to similar units (mg CO2-C m-2 h-1, µg CH4-C m-2 h-1 and µg N2O-N m-2 h-1). Based on Web of Knowledge search July 15th 2017 with the search terms "grassland", "pasture", "greenhouse gas", "ploughing" and/or "tillage". Only two studies representing conversion from pasture to cropland or other systems were included in this table.

| Publication | Grassland type | Observation Period | Measurement technique | CO2-C | CH4-C | N2O-N | Supporting Information |
|---|---|---|---|---|---|---|---|
| Bertora et al. 2007 | permanent pasture | 62 days | Incubation study of soil cores | 188 - 330 mg kg-1 soil * | NA | 50 - 1000 µg kg-1 soil * | Simulated ploughing, varying moisture contents, earthworm fertilizer application between 36 - 133 kg N ha-1 yr-1, conversion to cropland |
| Li et al. 2015 | managed grassland | approx five years grassland followed by three years of cropland | static GHG chamber | > 600 mg m-2 h-1 & | NA | > 1000 µg m-2 h-1 & | 15N gas flux method, restoration, |
| Buchen et al. 2016 | managed grassland | 44 days | 15N isotopic measurements | NA | NA | 100 - 1000 µg m-2 h-1 ^ | two soil types, conversion to two soil types, N2O emissions |
| Krol et al.2016 | permanent grassland | 17 weeks | static GHG chambers on lysimeter | NA | NA | 3000 µg m-2 h-1 % | and N leaching two adjacent fields (tilled and |
| Cowan et al. 2016 | permanent grassland | 175 days | eddy covariance | NA | NA | 500 - 700 µg m-2 h-1 $ | untilled) comparing ploughed and un- |
| Drewer et al. 2016 | permanent grassland poorly drained | three years | static GHG chambers/eddy covaria | 250 - 2000 mg m-2 h-1 $ | 1000 - 8000µg m-2 h-1 $ | 500 - 7000 µg m-2 h-1 $ | ploughed grassland |
| MacDonald et al. 2011 | grassland | | static GHG chambers | NA | NA | > 6000 µg m-2 h-1 ! | grassland converted to fallow |
| Estavillo et al. 2001 | permanent pasture | | incubation study of soil cores | NA | NA | 1800 - 5000 µg m-2 h-1 § | three treatments with different fertilizer levels, N2O and N2 conventional management with |
| Merbold et al. 2014 and this study | permanent grassland | five years | static GHG chambers/eddy covariance | > 400 mg m-2 h-1 # | non-different from zero | > 2000 µg m-2 h-1 # | restoration occuring after two years |

* cumulative fluxes over 62 days, & conversion from grassland to cropland, ^ approximate value recalculated from figure in the paper, % approximate peak emission following restoration calculated from figure in the paper, $ approximate value recalculated from figures presented in both paper, ! approximate value recalculated from figure in the paper, § approximate value presented in Figure 3 in the publication, # peak emissions

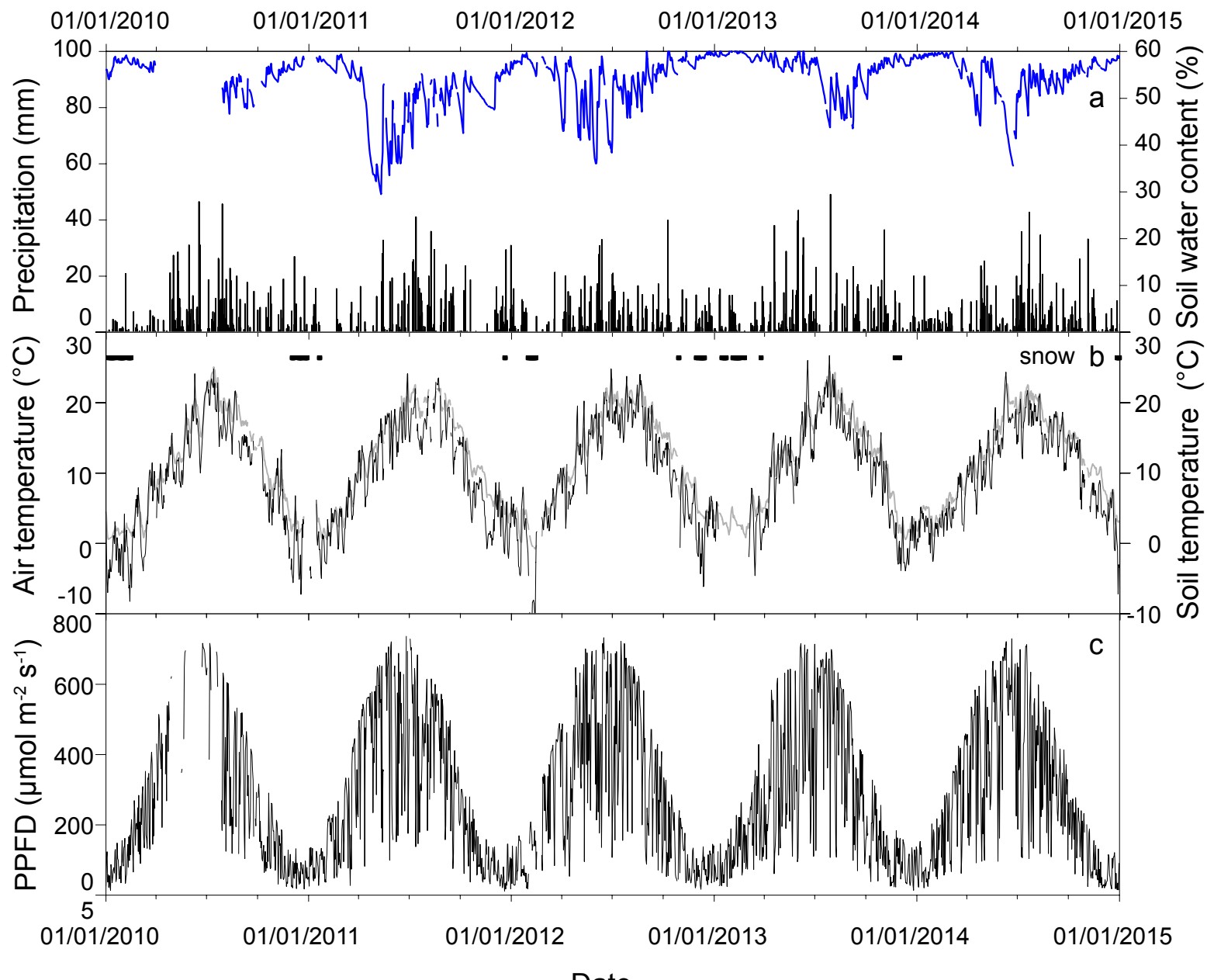

**Figure 1**: Weather conditions during the years 2010 – 2014. Weather data were measured with our meteorological sensors installed on site. (a) Daily sum of precipitation (mm) and soil water content (blue line, %) measured at 5 cm soil depth; (b) daily averaged air temperature (black line, °C), daily averaged soil temperature (grey line, °C) , and days with snow cover (horizontal bars); (c) daily averaged photosynthetic photon flux density (PPFD, µmol m⁻² s⁻¹). Snow covered days were identified with albedo calculations. Days with albedo values > 0.45 were identified as days with either snow or hoarfrost cover.

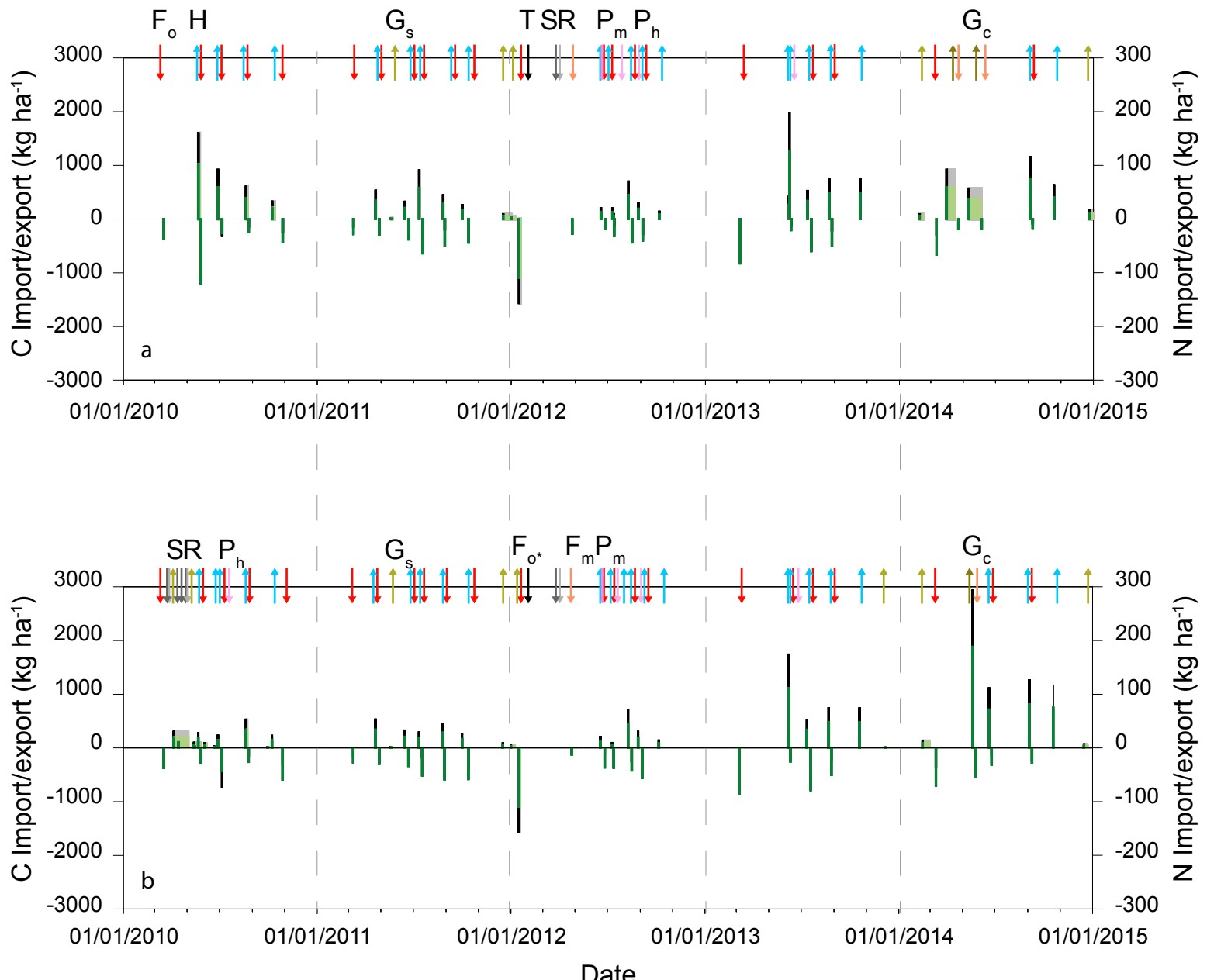

**Figure 2**: Management activities for both parcels (A and B in panels (a) and (b), respectively) on the CH-Cha site. Overall management varied particularly in 2010 between both parcels, whereas similar management took place between 2011 and 2014. Arrow direction indicates whether carbon (C in kg ha-1) and/or nitrogen (N in kg ha-1) were amended to, or exported from the site ("$F_o$" and "$F_{o*}$"- organic fertilizers, slurry/mannure (red); "$F_m$" - mineral fertilizer (light orange); "H" - harvest (light blue); "$G_s$" and "$G_c$" - grazing with sheep/cows (light/ dark brown). Other coloured arrows visualize any other management activities such as pesticide application ("$P_h$"- herbicide (light pink); "$P_m$"- molluscicide (dark pink); "T"- tillage (black), "R"- rolling (light grey) and "S"- sowing (dark grey) which occurred predominantly in 2010 (parcel B) and 2012 (parcels A and B). Carbon imports and exports are indicated by black and grey bars. Thereby black indicated the start of the specific management activities and grey the duration (e.g. during grazing, "Gs"). Green colors indicate nitrogen amendments or losses, with dark green visualizing the start of the activity and light green colors indicating the duration. Sign convention: positive values denote export/release, negative values import/uptake.

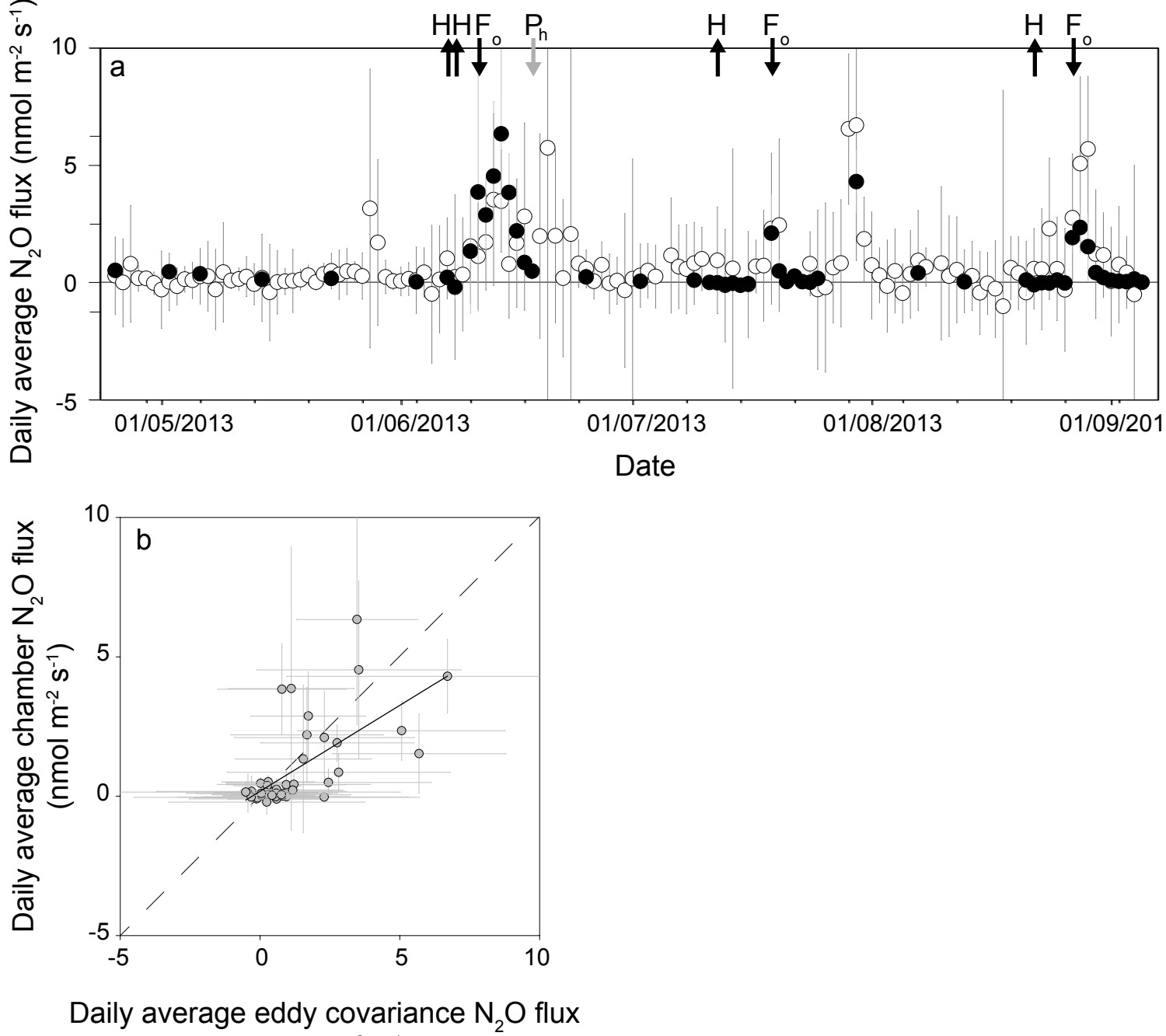

**Figure 3**: (a) Temporal dynamices of daily avergae N₂O fluxes measured with the eddy covariance (white circles) and manual greenhouse gas chambers (black circles) in 2013. Black arrows indicate management events, grey lines indicate standard deviation ("H"= harvest, "F$_o$" = organic fertilizer application (slurry), "P$_h$"= pesticide (herbicide) application);

(b) 1:1 comparison between chamber based and eddy covariance based N₂O fluxes in 2013. The dashed line represents the 1:1 line. (Regression: $y = 0.61x+0.17$, $r^2 = 0.4$). Sign convention: positive values denote export/release, negative values import/uptake.

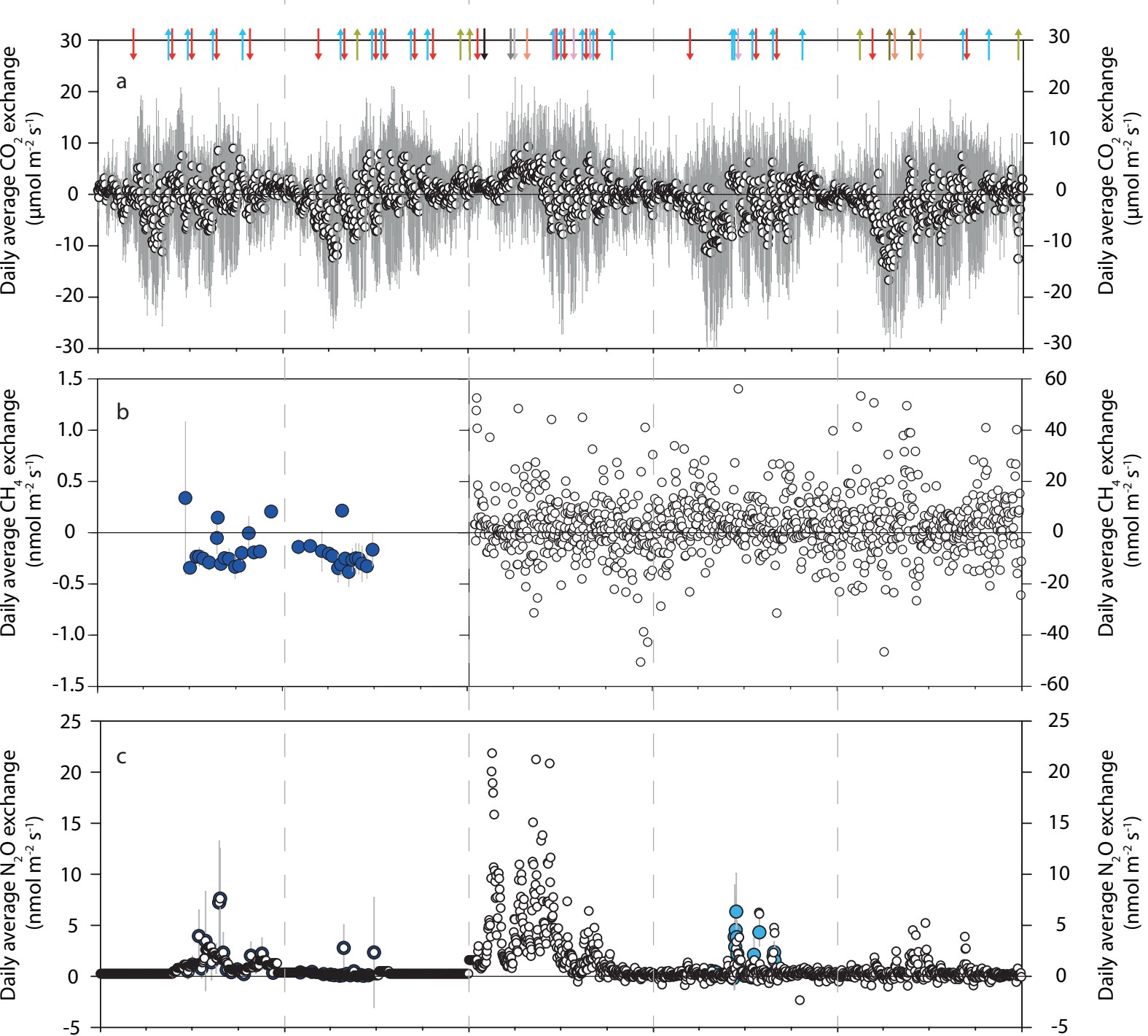

**Figure 4:** Temporal dynamics of gap-filled (except for CH4 in 200/2011) daily averaged greenhouse gas (GHG) fluxes (white circles):
a) (CO2 exchange in µmol m-2 s-1; (b) CH4 exchange in nmol m-2 s-1 and (c) N2O exchange in nmol m-2 s-1. Coloured circles indicate manual chamber measurements. While both GHGs, CH4 and N2O were measured in 2010 and 2011 (blue cirlces), N2O only was measured in 2013 (light blue circles). The grey dashed lines indicate the beginning of a new year. Same color coding as used in Figure 3 a was used to highlight management activities. Sign convention: positive values denote export/release, negative values import/uptake. Grey lines behind the circles indicate standard deviation.

**Table S1:** Detailed management information for the two parcels under investigation at the Chamau research station. Data are based on fieldbooks provided by the farm personnel as well as in-situ measurements. Organic fertilizer samples were sent to a central laboratory for nutrient content analysis (Labor fuer Boden- und Umweltanalytik, Eric Schweizer AG, Thun, Switzerland). Destructive harvests (n = 10) of biomass were carried out in the years 2010, 2011, 2013 and 2014. Harvest estimates are based on values derived from the in-situ measurements and data provided by the farm personnel. Detailed information on the grazing regime was furthermore provided by the farm personnel in hand-written form (not shown).

| Year | Mgmt. start (yyyy-mm-dd) | Mgmt. end (YYYY-MM-DD) | Regrowth | Mgmt. type | Characteristics | amt/ha | unit | N avail. (kg/ha) | $P_2O_5$ (kg/ha) | $K_2O$ (kg/ha) | Mg (kg/ha) | DM (%) | N total (g/kg DM) | N-$NH_4$ (g/kg DM) | C org (g/kg DM) | C/N | C (kg/ha) | N (kg/ha) |
|---|---|---|---|---|---|---|---|---|---|---|---|---|---|---|---|---|---|---|
| **2010** | | | | | | | | | | | | | | | | | | |
| | Parcel A, 1.4 ha | | | | | | | | | | | | | | | | | |
| | 2010-03-18 | | | organic fertilizer | slurry | 26 | m³ | 28.7 | 17.2 | 52.1 | 2.6 | 2.3 | 63.0 | 26.1 | 406.6 | 6.5 | 244 | 37.8 |
| | 2010-05-27 | | | organic fertilizer | slurry | 48 | m³ | 52.7 | 31.6 | 95.8 | 4.8 | 2.6 | 97.7 | 32.8 | 401.6 | 4.1 | 498 | 121.2 |
| | 2010-07-06 | | | organic fertilizer | slurry | 25 | m³ | 17.7 | 15.2 | 65.9 | 2.5 | 2.8 | 37.8 | 15.3 | 446.5 | 11.8 | 311 | 26.4 |
| | 2010-08-25 | | | organic fertilizer | slurry | 27 | m³ | 30.2 | 18.1 | 54.9 | 2.7 | 1.4 | 65.4 | 30.2 | 375.8 | 5.7 | 140 | 24.4 |
| | 2010-10-28 | | | organic fertilizer | slurry | 40 | m³ | 44.2 | 26.5 | 80.3 | 4.0 | 1.5 | 72.5 | 32.2 | 388.2 | 5.4 | 232 | 43.4 |
| | 2010-05-22 | 2010-05-24 | 1 | mowing | silage | 4 | waggon | | | | | 37.0 | | | | | 1598 | 102.8 |
| | 2010-06-28 | 2010-06-29 | 2 | mowing | small bales (pressed) | 99 | piece | | | | | 86.0 | | | | | 919 | 59.1 |
| | 2010-08-20 | 2010-08-22 | 3 | mowing | rowen | 1 | waggon | | | | | 70.0 | | | | | 605 | 38.9 |
| | 2010-10-08 | 2010-10-12 | 4 | mowing | silage bales | 4 | piece | | | | | 37.0 | | | | | 328 | 21.1 |
| | 2010-08-25 | | | | organically farmed land | | | | | | | | | | | | | |
| | 2010-09-29 | | | | organically farmed land | | | | | | | | | | | | | |
| amount of nutrient needed according to fertilization plan | | | | | | | | 130.0 | 80.0 | 240.0 | 30.0 | | | | | | | |
| balance of nutrient at the end of the year | | | | | | | 2 | -43.5 | -28.6 | -109.0 | 13.3 | | | | | | | 31 |
| | Parcel B, 3.3 ha | | | | | | | | | | | | | | | | | |
| | 2010-03-18 | | | organic fertilizer | slurry | 26 | m³ | 28.7 | 17.2 | 52.1 | 2.6 | 2.3 | 62.2 | 26.5 | 404.0 | 6.5 | 242 | 37.3 |
| | 2010-05-27 | | | organic fertilizer | slurry | 12 | m³ | 13.5 | 8.1 | 24.5 | 1.2 | 1.7 | 141.3 | 52.1 | 371.3 | 2.6 | 76 | 29.0 |
| | 2010-07-06 | | | organic fertilizer | slurry | 25 | m³ | 17.8 | 15.3 | 66.2 | 2.5 | 5.8 | 29.2 | 7.9 | 487.9 | 16.7 | 719 | 43.0 |
| | 2010-08-25 | | | organic fertilizer | slurry | 28 | m³ | 30.7 | 18.4 | 55.8 | 2.8 | 1.4 | 67.9 | 30.7 | 374.8 | 5.5 | 143 | 26.0 |
| | 2010-10-28 | | | organic fertilizer | slurry | 40 | m³ | 44.5 | 26.7 | 81.0 | 4.0 | 1.9 | 77.3 | 30.7 | 400.1 | 5.2 | 306 | 59.1 |
| | 2010-04-06 | 2010-05-03 | 1 | grazing | sheep | 18 | piece | | | | | | | | | | 305 | 19.6 |
| | 2010-04-15 | 2010-05-03 | 1 | grazing | sheep | 15 | piece | | | | | | | | | | 169 | 10.9 |
| | 2010-05-14 | 2010-05-27 | 2 | grazing | sheep | 12 | piece | | | | | | | | | | 98 | 6.3 |
| | 2010-05-15 | 2010-05-19 | 2 | grazing | sheep | 18 | piece | | | | | | | | | | 45 | 2.9 |
| | 2010-06-03 | 2010-06-06 | 2 | grazing | sheep | 49 | piece | | | | | | | | | | 90 | 5.8 |
| | 2010-06-21 | 2010-06-22 | 2 | grazing | sheep | 49 | piece | | | | | | | | | | 30 | 1.9 |
| | 2010-09-30 | 2010-10-01 | 4 | grazing | sheep | 31 | piece | | | | | | | | | | 19 | 1.2 |
| | 2010-05-22 | 2010-05-24 | 1 | mowing | silage | 1 | waggon | | | | | 37.0 | | | | | 278 | 17.9 |
| | 2010-06-28 | 2010-06-29 | 2 | mowing | small bales (pressed) | 25 | piece | | | | | 86.0 | | | | | 229 | 14.7 |
| | 2010-08-20 | 2010-08-22 | 3 | mowing | rowen | 1 | waggon | | | | | 70.0 | | | | | 527 | 33.9 |
| | 2010-10-08 | 2010-10-12 | 4 | mowing | silage bales | 2 | piece | | | | | 37.0 | | | | | 228 | 14.7 |
| | 2010-09-29 | | | | organically farmed land | | | | | | | | | | | | | |
| | 2010-07-14 | | herbicide | herbicide tabs (Ally) | | | | | | | | | | | | | | |
| | 2010-03-24 | | rolling | roller | | | | | | | | | | | | | | |
| | 2010-03-20 | | resowing | * OH-440 RenoTurbo | 7 | kg | | | | | | | | | | | | |
| | 2010-04-12 | | resowing | * OH-440 RenoTurbo | 9 | kg | | | | | | | | | | | | |
| | 2010-04-19 | | resowing | * OH 440 Reno | 6 | kg | | | | | | | | | | | | |
| | 2010-04-23 | | resowing | * OH-240 Reno | 3 | kg | | | | | | | | | | | | |
| | 2010-04-28 | | resowing | * OH-240 Reno | 3 | kg | | | | | | | | | | | | |
| amount of nutrient needed according to fertilization plan | | | | | | | | 130.0 | 80.0 | 240.0 | 30.0 | | | | | | | |
| balance of nutrient at the end of the year | | | | | | | | -5.2 | -5.7 | -39.7 | 16.8 | | | | | | | 64.5 |
| **2011** | | | | | | | | | | | | | | | | | | |
| | Parcel A, 1.5 ha | | | | | | | | | | | | | | | | | |
| | 2011-03-10 | | | organic fertilizer | slurry | 25.3 | m³ | 45.4 | 34.5 | 152.5 | 9.6 | 1.5 | 72.1 | 35.7 | 360.6 | 5.0 | 140.7 | 28.1 |
| | 2011-04-28 | | | organic fertilizer | slurry | 27.0 | m³ | 34.9 | 33.7 | 75.5 | 6.7 | 1.8 | 63.1 | 26.7 | 383.7 | 6.1 | 182.2 | 29.9 |
| | 2011-06-22 | | | organic fertilizer | slurry | 21.9 | m³ | 39.3 | 29.8 | 132.1 | 8.3 | 1.94 | 89.7 | 52.6 | 383.0 | 4.3 | 163.0 | 38.2 |
| | 2011-07-18 | | | organic fertilizer | slurry | 27.3 | m³ | 48.9 | 37.1 | 164.1 | 10.4 | 2.78 | 84.5 | 43.5 | 412.9 | 4.9 | 312.9 | 64.1 |
| | 2011-08-29 | | | organic fertilizer | slurry | 30.0 | m³ | 53.8 | 40.8 | 180.6 | 11.4 | 1.82 | 90.7 | 56.6 | 356.3 | 3.9 | 194.5 | 49.5 |
| | 2011-10-13 | | | organic fertilizer | slurry | 39.4 | m³ | 70.7 | 53.6 | 237.3 | 15.0 | 1.5 | 74.7 | 40.0 | 387.0 | 5.2 | 228.8 | 44.2 |
| | 2011-05-20 | 2011-05-22 | | grazing | sheep | 12.2 | piece | | | | | | | | | | 15.0 | 1.0 |
| | 2011-12-17 | 2011-12-31 | | grazing | sheep | 10.2 | piece | | | | | | | | | | 87.6 | 5.6 |
| | 2011-04-21 | 2011-04-21 | 1 | mowing | silage | 35.1 | dt DM | | | | | 37 | | | | | 530.7 | 34.1 |
| | 2011-06-15 | 2011-06-15 | 2 | mowing | silage | 21.4 | dt DM | | | | | 37 | | | | | 324.2 | 20.9 |
| | 2011-07-12 | 2011-07-12 | 3 | mowing | silage | 60.0 | dt DM | | | | | 37 | | | | | 907.8 | 58.4 |
| | 2011-08-26 | 2011-08-26 | 4 | mowing | rowen | 15.6 | dt DM | | | | | 70 | | | | | 445.4 | 28.6 |
| | 2011-10-01 | 2011-10-01 | 5 | mowing | silage | 17.2 | dt DM | | | | | 37 | | | | | 259.6 | 16.7 |
| amount of nutrient needed according to fertilization plan | | | | | | | | 0.0 | 80.0 | 240.0 | 30.0 | | | | | | | |
| balance of nutrient at the end of the year | | | | | | | | NA | NA | NA | NA | | | | | | | 88.7 |
| | Parcel B, 3.4ha | | | | | | | | | | | | | | | | | |
| | 2011-03-10 | | | organic fertilizer | slurry | 24.7 | m³ | 44.3 | 33.6 | 148.7 | 9.4 | 1.55 | 71.0 | 33.5 | 365.0 | 5.1 | 139.8 | 27.2 |
| | 2011-04-28 | | | organic fertilizer | slurry | 27.0 | m³ | 34.9 | 33.7 | 75.5 | 6.7 | 1.8 | 61.1 | 25.6 | 386.5 | 6.3 | 187.6 | 29.7 |
| | 2011-06-22 | | | organic fertilizer | slurry | 21.9 | m³ | 39.3 | 29.8 | 132.1 | 8.3 | 3.05 | 50.5 | 32.5 | 436.6 | 8.7 | 292.1 | 33.8 |
| | 2011-07-18 | | | organic fertilizer | slurry | 27.3 | m³ | 48.9 | 37.1 | 164.1 | 10.4 | 1.76 | 106.8 | 59.1 | 375.5 | 3.5 | 180.2 | 51.2 |
| | 2011-08-29 | | | organic fertilizer | slurry | 30.0 | m³ | 53.8 | 40.8 | 180.6 | 11.4 | 2.57 | 75.9 | 43.6 | 399.4 | 5.3 | 308.0 | 58.5 |
| | 2011-10-13 | | | organic fertilizer | slurry | 39.4 | m³ | 70.7 | 53.6 | 237.3 | 15.0 | 2.79 | 52.7 | 23.3 | 365.7 | 6.9 | 402.2 | 57.9 |
| | 2011-05-20 | 2011-05-22 | | grazing | sheep | 12.2 | piece | | | | | | | | | | 15.0 | 1.0 |
| | 2011-12-17 | 2011-12-31 | | grazing | sheep | 10.2 | piece | | | | | | | | | | 87.6 | 5.6 |
| | 2011-04-21 | 2011-04-21 | 1 | mowing | silage | 35.1 | dt DM | | | | | 37 | | | | | 530.8 | 34.1 |
| | 2011-06-15 | 2011-06-15 | 2 | mowing | silage | 21.4 | dt DM | | | | | 37 | | | | | 324.2 | 20.9 |
| | 2011-07-12 | 2011-07-12 | 3 | mowing | silage | 19.1 | dt DM | | | | | 37 | | | | | 289.7 | 18.6 |
| | 2011-08-26 | 2011-08-26 | 4 | mowing | rowen | 15.6 | dt DM | | | | | 70 | | | | | 445.4 | 28.6 |
| | 2011-10-01 | 2011-10-01 | 5 | mowing | silage | 17.2 | dt DM | | | | | 37 | | | | | 259.6 | 16.7 |
| amount of nutrient needed according to fertilization plan | | | | | | | | 0.0 | 80.0 | 240.0 | 30.0 | | | | | | | |
| balance of nutrient at the end of the year | | | | | | | | 291.9 | 228.6 | 938.3 | 61.2 | | | | | | | 132.8 |
| **2012** | | | | | | | | | | | | | | | | | | |
| | Parcel A, 1.5 ha | | | | | | | | | | | | | | | | | |
| | 2012-03-28 | | | harrowing | harrow | | | | | | | | | | | | | |
| | 2012-02-02 | | | ploughing | plough | | | | | | | | | | | | | |
| | 2012-03-29 | | | rolling | roller | | | | | | | | | | | | | |
| | 2012-04-25 | | | mineral fertilizer | **calcium carbonate & ammonium nitrate | 100.0 | kg | 27.5 | 0.0 | 0.0 | 0.0 | 100 | 275.0 | 270.0 | 0.0 | 0.0 | 0.0 | 27.5 |
| | 2012-01-16 | 2012-01-18 | | organic fertilizer | manure | 20.4 | t | 27.5 | 46.9 | 173.3 | 16.9 | 16.71 | 32.1 | 3.6 | 458.6 | 14.3 | 1563.9 | 109.4 |
| | 2012-06-26 | | | organic fertilizer | slurry | 18.8 | m³ | 29.1 | 20.7 | 52.6 | 4.7 | 0.98 | 100.2 | 69.5 | 321.7 | 3.2 | 59.2 | 18.4 |
| | 2012-07-13 | | | organic fertilizer | slurry | 18.8 | m³ | 29.1 | 20.7 | 52.6 | 4.7 | 1.82 | 92.9 | 46.2 | 372.0 | 4.0 | 127.1 | 31.7 |
| | 2012-08-16 | | | organic fertilizer | slurry | 19.6 | m³ | 30.4 | 21.6 | 54.9 | 4.9 | 2.81 | 79.8 | 40.3 | 381.2 | 4.8 | 209.8 | 43.9 |
| | 2012-09-05 | | | organic fertilizer | slurry | 19.6 | m³ | 30.4 | 21.6 | 54.9 | 4.9 | 3.55 | 57.8 | 31.0 | 406.3 | 7.0 | 282.5 | 40.2 |
| | 2012-01-01 | 2012-01-08 | | grazing | sheep | 10.2 | piece | | | | | | | | | | 43.8 | 2.8 |
| | 2012-06-18 | 2012-06-18 | 1 | mowing | silage | 13.5 | dt DM | | | | | 37 | | | | | 203.7 | 13.1 |

| Start date | End date | No. | Activity | Product | Amount | Unit | c1 | c2 | c3 | c4 | c5 | c6 | c7 | c8 | c9 | c10 | c11 |
|---|---|---|---|---|---|---|---|---|---|---|---|---|---|---|---|---|---|
| 2012-07-10 | 2012-07-10 | 2 | mowing | silage | 13.5 | dt DM | | | | | | 37 | | | | 203.7 | 13.1 |
| 2012-07-12 | 2012-07-12 | 2 | mowing | silage | 6.1 | dt DM | | | | | | 37 | | | | 92.6 | 6.0 |
| 2012-08-09 | 2012-08-09 | 3 | mowing | rowen | 24.5 | dt DM | | | | | | 70 | | | | 700.9 | 45.1 |
| 2012-08-28 | 2012-08-28 | 4 | mowing | silage | 19.9 | dt DM | | | | | | 37 | | | | 301.1 | 19.4 |
| 2012-10-06 | 2012-10-06 | 5 | mowing | silage | 9.2 | dt DM | | | | | | 37 | | | | 139.0 | 8.9 |
| 2012-07-17 | | | herbicide | *** Harmony tablets | 2.7 | tablets | | | | | | | | | | | |
| 2012-09-06 | | | herbicide | **** Asulam | 4.0 | l | | | | | | | | | | | |
| 2012-06-19 | | | pesticide | ***** snail bait | 7.0 | kg | | | | | | | | | | | |
| 2012-03-28 | | | resowing | * OH-440 Extra | 32.0 | kg | | | | | | | | | | | |
| amount of nutrient needed according to fertilization plan | | | | | | | 0.0 | 96.0 | 288.0 | 36.0 | | | | | | | |
| balance of nutrient at the end of the year | | | | | | | NA | NA | NA | NA | | | | | | | 162.8 |
| **Parcel B, 3.4ha** | | | | | | | | | | | | | | | | | |
| 2012-03-28 | | | harrowing | harrow | | | | | | | | | | | | | |
| 2012-02-02 | | | ploughing | plough | | | | | | | | | | | | | |
| 2012-03-29 | | | rolling | roller | | | | | | | | | | | | | |
| 2012-04-25 | | | mineral fertilizer | **calcium carbonate & ammonium nitrate | 45.0 | kg | 27.5 | 0.0 | 0.0 | 0.0 | 100 | 275.0 | 270.0 | 0.0 | 0.0 | 0.0 | 12.4 |
| 2012-01-16 | 2012-01-18 | | organic fertilizer | manure | 20.4 | t | 27.5 | 46.9 | 173.3 | 16.9 | 16.71 | 32.1 | 3.6 | 458.6 | 14.3 | 1564.0 | 109.4 |
| 2012-06-26 | | | organic fertilizer | slurry | 18.8 | m³ | 29.1 | 20.7 | 52.6 | 4.7 | 1.55 | 125.7 | 60.6 | 367.2 | 2.9 | 106.9 | 36.6 |
| 2012-07-13 | | | organic fertilizer | slurry | 18.8 | m³ | 29.1 | 20.7 | 52.6 | 4.7 | 1.65 | 119.2 | 54.5 | 371.4 | 3.1 | 115.1 | 36.9 |
| 2012-08-16 | | | organic fertilizer | slurry | 19.6 | m³ | 30.4 | 21.6 | 54.9 | 4.9 | 3.13 | 68.6 | 38.3 | 400.8 | 5.8 | 245.8 | 42.1 |
| 2012-09-05 | | | organic fertilizer | slurry | 19.6 | m³ | 30.4 | 21.6 | 54.9 | 4.9 | 2.52 | 113.1 | 52.4 | 399.5 | 3.5 | 197.3 | 55.8 |
| 2012-01-01 | 2012-01-08 | | grazing | sheep | 10.2 | piece | | | | | | | | | | 43.8 | 2.8 |
| 2012-06-18 | 2012-06-18 | 1 | mowing | silage | 13.5 | dt DM | | | | | | 37 | | | | 203.8 | 13.1 |
| 2012-07-10 | 2012-07-10 | 2 | mowing | silage | 6.1 | dt DM | | | | | | 37 | | | | 92.6 | 6.0 |
| 2012-08-09 | 2012-08-09 | 3 | mowing | rowen | 24.5 | dt DM | | | | | | 70 | | | | 701.0 | 45.1 |
| 2012-08-28 | 2012-08-28 | 4 | mowing | silage | 19.9 | dt DM | | | | | | 37 | | | | 301.0 | 19.4 |
| 2012-10-06 | 2012-10-06 | 5 | mowing | silage | 9.2 | dt DM | | | | | | 37 | | | | 138.9 | 8.9 |
| 2012-07-17 | | | herbicide | *** Harmony tablets | 1.2 | tablets | | | | | | | | | | | |
| 2012-09-06 | | | herbicide | **** Asulam | 4.0 | l | | | | | | | | | | | |
| 2012-06-19 | | | pesticide | ***** snail bait | 7.0 | kg | | | | | | | | | | | |
| 2012-03-28 | | | resowing | * OH-440 Extra | 32.0 | kg | | | | | | | | | | | |
| amount of nutrient needed according to fertilization plan | | | | | | | 0.0 | 96.0 | 288.0 | 36.0 | | | | | | | |
| balance of nutrient at the end of the year | | | | | | | NA | NA | NA | NA | | | | | | | 197.9 |
| **2013** | | | | | | | | | | | | | | | | | |
| **Parcel A, 1.5ha** | | | | | | | | | | | | | | | | | |
| 2013-03-07 | | | organic fertilizer | slurry | 24.5 | m³ | 38.0 | 26.9 | 68.6 | 6.1 | 3.5 | 96.7 | 69.3 | 397.2 | 4.1 | 340.5 | 82.9 |
| 2013-06-11 | | | organic fertilizer | slurry | 20.0 | m³ | 31.1 | 22.0 | 56.0 | 5.0 | 1.7 | 61.2 | 37.6 | 378.3 | 6.2 | 128.6 | 20.8 |
| 2013-07-19 | | | organic fertilizer | slurry | 26.7 | m³ | 41.5 | 29.4 | 74.9 | 6.7 | 2.4 | 93.6 | 59.4 | 377.4 | 4.0 | 242.1 | 60.0 |
| 2013-08-27 | | | organic fertilizer | slurry | 29.0 | m³ | 45.0 | 31.9 | 81.1 | 7.2 | 1.9 | 89.8 | 56.2 | 391.6 | 4.4 | 215.6 | 49.4 |
| 2013-06-07 | 2013-06-07 | 1 | mowing | silage | 27.6 | dt DM | | | | | | 37 | | | | 416.9 | 26.8 |
| 2013-06-08 | 2013-06-08 | 1 | mowing | hay | 56.0 | dt DM | | | | | | 37 | | | | 847.2 | 54.5 |
| 2013-07-12 | 2013-07-12 | 2 | mowing | rowen | 18.4 | dt DM | | | | | | 70 | | | | 525.7 | 33.8 |
| 2013-08-22 | 2013-08-22 | 3 | mowing | silage | 49.0 | dt DM | | | | | | 37 | | | | 741.0 | 47.7 |
| 2013-10-19 | 2013-10-19 | 4 | mowing | silage | 49.0 | dt DM | | | | | | 37 | | | | 741.0 | 47.7 |
| 2013-06-18 | | | herbicide | *** Harmony tablets | 2.0 | tablets | | | | | | | | | | | |
| amount of nutrient needed according to fertilization plan | | | | | | | 0.0 | 96.0 | 288.0 | 36.0 | | | | | | | |
| balance of nutrient at the end of the year | | | | | | | 155.6 | 110.2 | 280.6 | 25.1 | | | | | | | |
| **Parcel B, 3.4ha** | | | | | | | | | | | | | | | | | |
| 2013-03-07 | | | organic fertilizer | slurry | 24.5 | m³ | 38.0 | 26.9 | 68.6 | 6.1 | 3.3 | 106.0 | 74.3 | 396.2 | 3.7 | 320.2 | 85.7 |
| 2013-06-11 | | | organic fertilizer | slurry | 20.0 | m³ | 31.0 | 22.0 | 56.0 | 5.0 | 2 | 62.9 | 37.7 | 390.9 | 6.2 | 156.4 | 25.2 |
| 2013-07-19 | | | organic fertilizer | slurry | 26.7 | m³ | 41.5 | 29.4 | 74.9 | 6.7 | 2.8 | 105.0 | 59.0 | 401.8 | 3.8 | 300.8 | 78.6 |
| 2013-08-27 | | | organic fertilizer | slurry | 29.0 | m³ | 45.0 | 31.9 | 81.1 | 7.2 | 2 | 87.2 | 51.3 | 386.0 | 4.4 | 223.7 | 50.5 |
| 2013-12-06 | 2013-12-07 | | grazing | sheep | 20.6 | piece | | | | | | | | | | 12.6 | 0.8 |
| 2013-06-07 | | 1 | mowing | silage | 27.5 | dt DM | | | | | | 37 | | | | 416.8 | 26.8 |
| 2013-06-08 | | 1 | mowing | hay | 49.4 | dt DM | | | | | | 86 | | | | 1737.6 | 111.8 |
| 2013-07-12 | | 2 | mowing | rowen | 18.4 | dt DM | | | | | | 70 | | | | 525.7 | 33.8 |
| 2013-08-22 | | 3 | mowing | silage | 49.0 | dt DM | | | | | | 37 | | | | 741.0 | 47.7 |
| 2013-10-19 | | 4 | mowing | silage | 49.0 | dt DM | | | | | | 37 | | | | 741.0 | 47.7 |
| 2013-06-18 | | | herbicide | *** Harmony tablets | 2.0 | tablets | | | | | | | | | | | |
| amount of nutrient needed according to fertilization plan | | | | | | | 0.0 | 96.0 | 288.0 | 36.0 | | | | | | | |
| balance of nutrient at the end of the year | | | | | | | 155.6 | 110.2 | 280.6 | 25.1 | | | | | | | -28.5 |
| **2014** | | | | | | | | | | | | | | | | | |
| **Parcel A, 2.2ha** | | | | | | | | | | | | | | | | | |
| 2014-04-22 | | | mineral fertilizer | **calcium carbonate & ammonium nitrate | 68.2 | kg | 18.8 | 0.0 | 0.0 | 0.0 | 100 | 275.0 | 270.0 | 0.0 | 0.0 | 0.0 | 18.8 |
| 2014-06-05 | | | mineral fertilizer | **calcium carbonate & ammonium nitrate | 68.2 | kg | 18.8 | 0.0 | 0.0 | 0.0 | 100 | 275.0 | 275.0 | 0.0 | 0.0 | 0.0 | 18.8 |
| 2014-03-12 | | | organic fertilizer | slurry | 27.3 | m³ | 54.9 | 32.7 | 76.4 | 6.8 | 2.8 | 87.2 | 56.1 | 394.6 | 4.5 | 301.3 | 66.6 |
| 2014-09-09 | | | organic fertilizer | slurry | 28.2 | m³ | 56.7 | 33.8 | 78.9 | 7.0 | 0.8 | 79.8 | 44.9 | 371.3 | 4.7 | 83.7 | 18.0 |
| 2014-02-08 | 2014-02-15 | | grazing | sheep | 20.9 | piece | | | | | | | | | | 89.8 | 5.8 |
| 2014-03-31 | 2014-04-15 | 1 | grazing | cattle | 12.5 | piece | | | | | | | | | | 917.9 | 59.0 |
| 2014-05-12 | 2014-06-03 | 2 | grazing | cattle | 5.3 | piece | | | | | | | | | | 598.3 | 16.3 |
| 2014-09-04 | | 3 | mowing | silage | 76.4 | dt DM | | | | | | 37 | | | | 1233.6 | 41.1 |
| 2014-10-19 | | 4 | mowing | silage | 41.7 | dt DM | | | | | | 37 | | | | 662.6 | 36.9 |
| 2014-12-24 | 2015-01-07 | 5 | grazing | sheep | 19.1 | piece | | | | | | | | | | 163.9 | 10.5 |
| amount of nutrient needed according to fertilization plan | | | | | | | 0.0 | 96.0 | 288.0 | 36.0 | | | | | | | |
| balance of nutrient at the end of the year | | | | | | | 149.1 | 66.5 | 155.3 | 13.9 | | | | | | | -47.5 |
| **Parcel B, 2.7ha** | | | | | | | | | | | | | | | | | |
| 2014-03-12 | | | organic fertilizer | slurry | 28.1 | m³ | 56.6 | 33.8 | 78.8 | 7.0 | 3.5 | 71.6 | 44.5 | 367.9 | 5.1 | 362.4 | 70.5 |
| 2014-05-26 | | | organic fertilizer | slurry | 25.9 | m³ | 52.2 | 31.1 | 72.6 | 6.5 | 2.7 | 76.9 | 45.7 | 394.8 | 5.1 | 276.4 | 53.8 |
| 2014-06-25 | | | organic fertilizer | slurry | 20.7 | m³ | 41.7 | 24.9 | 58.1 | 5.2 | 1.6 | 93.7 | 55.6 | 377.3 | 4.0 | 125.2 | 31.1 |
| 2014-09-09 | | | organic fertilizer | slurry | 31.1 | m³ | 62.6 | 37.3 | 87.1 | 7.8 | 1.8 | 49.6 | 23.5 | 415.6 | 8.4 | 232.7 | 27.8 |
| 2014-02-15 | 2014-02-18 | | grazing | sheep | 17.0 | piece | | | | | | | | | | 31.3 | 2.0 |
| 2014-12-16 | 2014-12-23 | | grazing | sheep | 15.6 | piece | | | | | | | | | | 66.8 | 4.3 |
| 2014-05-20 | | 1 | mowing | hay | 83.3 | dt DM | | | | | | 86 | | | | 3117.8 | 91.3 |
| 2014-06-20 | | 2 | mowing | rowen | 38.9 | dt DM | | | | | | 70 | | | | 1173.7 | 56.2 |
| 2014-09-04 | | 3 | mowing | silage | 83.3 | dt DM | | | | | | 37 | | | | 1342.9 | 54.8 |
| 2014-10-19 | | 4 | mowing | silage | 77.1 | dt DM | | | | | | 37 | | | | 1237.6 | 67.4 |
| amount of nutrient needed according to fertilization plan | | | | | | | 0.0 | 96.0 | 288.0 | 36.0 | | | | | | | |
| balance of nutrient at the end of the year | | | | | | | 213.2 | 127.1 | 296.6 | 26.5 | | | | | | | -92.8 |

*Grass-white clover mixture 'OH 440 Extra/OH 240' (Otto Hauenstein Samen AG) consisting of red clover (Trifolium pratensis), white clover (Trifolium repens), english ryegrass (Lolium perenne), common meadow grass (Poa pratensis), red fescue (Festuca rubra) and common timothy (Phleum pratense).
**Calcium ammonium nitrate fertilizer - 13.5% nitrate, 13.5% ammonium (Flora Duengemittel, Hanweller, Saar, Germany).
*** Harmony tablet, 12 % Thifensulfuron-Methyl (Bayer AG, CropScience, 3052, Zollikofen, Switzerland).
****ASULAM (Agro Seller Discount AG, St. Gallen, Switzerland).
*****Snail bait (Steiner Ultra W-4935, Omya AG, Switzerland).