# Peer review of "Memory effects on greenhouse gas emissions ($CO_2$, $N_2O$ and $CH_4$) following"

_Biogeosciences, 2020_

## Referee Comment (RC1) · Anonymous Referee #2 · 22 Jun 2020

The study presented here title "Memory effects on greenhouse gas emissions (CO2, N2O and CH4) following grassland renovation?" presents trace gas measurements from 5 years of a grazed and harvest pasture in Switzerland including a pasture restoration event. In general, this is a well written and worthwhile study. Few studies report all greenhouse gases, and even fewer for multiple years and covering infrequent management activities. I believe this to be of publication quality following consideration of my commentary below. I have separated my comments into major, moderate and minor/technical concerns based on importance and impact to the manuscript as I see it. I believe these can be dealt with by the authors and would further enhance the manuscript.

[Figure]

Major concerns

1. CH4 fluxes: I have major concerns with the usage of the CH4 fluxes as presented in this manuscript. Firstly, while the authors present a comparison of N2O chamber and eddy covariance data (Figure 3), they do not for CH4. I believe this is likely as the comparison does not suggest any 1:1 relationship (based on my interpretation of Figure 4b). The authors then use this chamber data to derive annual CH4 fluxes for the years without EC data and assume to be comparable with the EC derived annual fluxes. From the data presented, I see no evidence to believe this to be the case (unlike N2O). Given the two chamber years suggest a small uptake of CH4, while the last three a release of CH4 coinciding with a difference in measurement methodology, I question whether the authors really believe these years are comparable. While the authors discuss these methodology differences in detail in the discussion section, and overall the contribution of CH4 to the GHG budget is small, I believe further attention needs to be given to this, and ideally the equivalent plot to figure 3b is presented for CH4. Based on the timing of management events (pasture restoration) and change in measurement methodology it could be easily interpreted as pasture restoration changes grassland CH4 exchange from an uptake to release.

2. The impact of grazing needs further consideration. While harvesting is more common in this study, the impact of grazing needs further clarification and/or modification of the presented results. Firstly, it is unclear to me how the grazing off-take was estimated (please clarify), and whether the deposition of excreta C was included in the C balances. While I'm not familiar with sheep grazing, at least for cattle this can be in the order of one-third of consumption, and therefore not an insignificant component (especially for 2014, Parcel A with 1769.9 kg C ha-1 of grazing removal according to table S1) and requiring acknowledgement of how this is currently dealt with, or included in the C balance (e.g. Table 2). Furthermore, the authors state they did not detect any CH4 release with grazing (lines 432-433). Using the example of Parcel A in 2014, which was primarily grazed by cattle, and assuming ∼3% was converted and released

as CH4 (e.g. Felber et al. (2016)), 53.1 kg C ha-1 would have been emitted from the grazers as CH4, which when converted to g CO2-eq m-2 calculated to 240 g CO2-eq m-2 or much larger than the 55 g CO2-eq m-2 reported in table. If this was not detected, then I suggest the authors reconsider how grazing related CH4 is dealt with in this manuscript given they are reporting ecosystem scale GHG budgets.

Moderate Concerns

3. The focus (or perhaps title?) of this manuscript needs sharpening. The title indicates a focus on pasture restoration which is matched by the abstract, yet much attention is given to methodological considerations. Specific goal (ii) states "briefly compare two different measurement techniques" however the first two-thirds of the discussion (i.e. not briefly) comments on this aspect! While important and noteworthy, either change the title/abstract, or return the primary focus of the discussion to management effects. Additionally, goal (iii) is not really explored in this manuscript – perhaps combine with goal (i)?

4. Providing a partial N budget provides little useful information. Including individual components is beneficial, but to sum them up as an incomplete "budget" is not. If the authors choose to retain the N budget, please include some further context including some ballpark estimates of the remaining components to aid interpretation.

5. While N2O flux gap filling is difficult, the use of running medians may be problematic, and especially for gaps occurring during pulse emissions (e.g. the restoration period/fertiliser applications). The authors should comment on limitations of this approach, especially in the absence of any uncertainties (which I accept is rarely done in N2O flux studies so do not see them as a requirement here).

Minor/Technical Concerns

Lines 33-34: grazing is listed as both a regular and sporadic management activity. Please clarify which it is.

Line 37: Missing the word "out" (or similar) after "carried".

Lines 86-89: Why did you hypothesis continuous losses of CO2? Several studies (e.g. (Rutledge et al., 2017; Ammann et al., 2020, etc) show CO2 uptake in restoration and later years.

Lines 89-90: If you expect CO2 losses (as per the above point), why would you expect a C gain? Please adjust this and align with the previous sentence to clarify your hypothesis.

Line 108: Do you mean CH4 emissions from the land or the grazers? In fact, this point needs clarity throughout the manuscript – are the grazers included within the system boundary, and therefore their emissions?

Lines 123-127: this sentence is very clunky – suggest reviewing.

Line 130: "adaptations" should be "adaptation" (no "s").

Line 137: "respectively" is not needed – please delete.

Lines 232-234: If an LI-7500 (rather than LI-7500A) was the self-heating correction applied?

Lines 241-249: It was unclear to me what QA/QC procedures were applied to the raw (10/20Hz) and which to the 30-minute data. I suggest improving the clarity here.

Line 248: what was considered the physically plausible range? Please include this information.

Line 280: Order of words: "no longer closed" should be "closed no longer".

Line 314: Remove the word "Up"

Line 413: Insert the word "and" between "(Figure 1c)" and "temperatures".

Lines 477-478: I think the before and after restoration periods should be separated. I don't believe averaging the two periods to be fair as part of the purpose of restoration is to improve growth, and therefore modification of $CO_2$ exchange should also be expected.

Line 480: According to Table 2, $CH_4$ emissions for 2013 and 2014 were actually >1 – please correct.

Line 538: Correct the format of the reference

Line 579-580: Are you referring to the measured $CO_2$ exchange to be $\pm 50$ g C m-2 y-1, or the uncertainty? This sentence is very unclear as no uncertainty has been presented, so please clarify.

Table 1: I find the "max data availability" columns repetitive – perhaps just a single column of this data?

Table 4: I suspect the labelling of Parcels A and B for both fertilizer and harvest are not correct. As written, fertilizer was only applied to Parcel A, and Harvest to Parcel B. Please correct is appropriate.

References

Ammann, C., Neftel, A., Jocher, M., Fuhrer, J., Leifeld, J., 2020. Effect of management and weather variations on the greenhouse gas budget of two grasslands during a 10-year experiment. Agric. Ecosyst. Environ. 292.

Felber, R., Bretscher, D., Münger, A., Neftel, A., Ammann, C., 2016. Determination of the carbon budget of a pasture: effect of system boundaries and flux uncertainties. Biogeosciences 13, 2959-2969.

Rutledge, S., Wall, A.M., Mudge, P.L., Troughton, B., Campbell, D.I., Pronger, J., Joshi, C., Schipper, L.A., 2017. The carbon balance of temperate grasslands part II: The impact of pasture renewal via direct drilling. Agric. Ecosyst. Environ. 239, 132-142.

---

## Referee Comment (RC2) · Anonymous Referee #1 · 24 Jun 2020

The technical revisions made by the authors are sufficient to progress with the manuscript.

I am unable to comment further until the data analysis of the fully revised version is complete as suggested by the authors.

---

## Author Comment (AC1) · 19 Aug 2020

**Response to reviewers on our manuscript "Memory effects on greenhouse gas emissions (CO2, N2O and CH4) following grassland restoration?" by Lutz Merbold et al.**

We thank both reviewers for their critical assessment and provide an answer on how we foresee to address the individual comments in a revised manuscript. Given that reviewer #1 already provided a full assessment during the access review phase we just want to restate again that we intend to provide the necessary changes in the revised manuscript as originally suggested with further amendments following the comments provided by reviewer #2 whom has raised has raised similar concerns and some others. Kindly find our response below. The reviewer's comment is stated first, followed by our response in *italic* font.

Reviewer #2:
The study presented here title "Memory effects on greenhouse gas emissions (CO2, N2O and CH4) following grassland renovation?" presents trace gas measurements from 5 years of a grazed and harvest pasture in Switzerland including a pasture restoration event. In general, this is a well written and worthwhile study. Few studies report all greenhouse gases, and even fewer for multiple years and covering infrequent management activities. I believe this to be of publication quality following consideration of my commentary below. I have separated my comments into major, moderate and minor/technical concerns based on importance and impact to the manuscript as I see it. I believe these can be dealt with by the authors and would further enhance the manuscript.

*We thank the reviewer for this positive assessment and share the opinion of few studies reporting on multiple years of GHG exchange measurements of the three GHGs covering specific management activities.*

Major concerns
1. CH4 fluxes: I have major concerns with the usage of the CH4 fluxes as presented in this manuscript. Firstly, while the authors present a comparison of N2O chamber and eddy covariance data (Figure 3), they do not for CH4. I believe this is likely as the comparison does not suggest any 1:1 relationship (based on my interpretation of Figure 4b). The authors then use this chamber data to derive annual CH4 fluxes for the years without EC data and assume to be comparable with the EC derived annual fluxes. From the data presented, I see no evidence to believe this to be the case (unlike N2O). Given the two chamber years suggest a small uptake of CH4, while the last three a release of CH4 coinciding with a difference in measurement methodology, I question whether the authors really believe these years are comparable. While the authors discuss these methodology differences in detail in the discussion section, and overall the contribution of CH4 to the GHG budget is small, I believe further attention needs to be given to this, and ideally the equivalent plot to figure 3b is presented for CH4. Based on the timing of management events (pasture restoration) and change in measurement methodology it could be easily interpreted as pasture restoration changes grassland CH4 exchange from an uptake to release.

*These are indeed relevant points and surely, we do not want to give the impression that pasture restoration changes grassland CH4 exchange from an uptake to release as this can not be proven by the data presented in this study (see following response). We had preferred to show a similar comparison as given for N2O, however the methane concentrations measurements were not reliable in 2013 due to a flame ionization detector (FID) malfunction in the gas chromatograph.*
*Overall, we also did not expect to find a similar relation between the methane flux measurements obtained by eddy covariance and chambers caused by the small magnitude of the fluxes measured. As stated in the original manuscript* "We calculated detection limits for the individual GHGs from our manual chambers following (Parkin et al., 2012). Detection limits were $0.34 \pm 0.26$ nmol m$^{-2}$ s$^{-1}$, $0.05 \pm 0.02$ nmol m$^{-2}$ s$^{-1}$, and $0.06 \pm 0.06$ μmol m$^{-2}$ s$^{-1}$ for CH$_4$, N$_2$O and CO$_2$, respectively, clearly indicating that methane fluxes measured by GHG

chambers in 2010/2011 were on average $-0.16 \pm 0.16$ nmol $CH_4$ m$^{-2}$ s$^{-1}$, (see Table 2) and thus below the actual detection limit."

*However, we did compare our eddy covariance methane flux values (methane fluxes fluctuating around 0 with an overall range of -40 up to +40 nmol CH4 m-2 s-1 (Figure 4 b)) with the values reported by (Felber et al., 2015) from a similar grassland system in Western Switzerland. (Felber et al., 2015) have shown that such values measured by the EC technique represent a soil signal (Figure 6 in Felber et al. 2015).*

*Following this, we agree that we should not have computed annual sums for the years 2010/2011 for methane and will remove these in the revised manuscript. Yet, we will remain with the numbers presented for methane in 2012 -2014. We want to stress again, that methane fluxes are of minor importance for the carbon and greenhouse gas budget of the site under the current management (see also our response to the following concern made by reviewer #2 on the influence of grazing animals on methane fluxes).*

2. The impact of grazing needs further consideration. While harvesting is more common in this study, the impact of grazing needs further clarification and/or modification of the presented results. Firstly, it is unclear to me how the grazing off-take was estimated (please clarify), and whether the deposition of excreta C was included in the C balances. While I'm not familiar with sheep grazing, at least for cattle this can be in the order of one-third of consumption, and therefore not an insignificant component (especially for 2014, Parcel A with 1769.9 kg C ha-1 of grazing removal according to table S1) and requiring acknowledgement of how this is currently dealt with, or included in the C balance (e.g. Table 2).

Furthermore, the authors state they did not detect any CH4 release with grazing (lines 432-433). Using the example of Parcel A in 2014, which was primarily grazed by cattle, and assuming _3% was converted and released as CH4 (e.g. Felber et al. (2016)), 53.1 kg C ha-1 would have been emitted from the grazers as CH4, which when converted to g CO2-eq m-2 calculated to 240 g CO2-eq m-2 or much larger than the 55 g CO2-eq m-2 reported in table. If this was not detected, then I suggest the authors reconsider how grazing related CH4 is dealt with in this manuscript given they are reporting ecosystem scale GHG budgets.

*Indeed, methane emissions from grazing animals need to be considered in annual budgets of methane and carbon. We argue that these are already accounted for in our data. What needs to be noted is that grazing intensity was extremely low and only lasted for few days in the specific years (2010, 2011, 2014). Also, most of the grazing were sheep, and cattle were only present in 2014 in Parcel A for less than four weeks in total at an average stocking rate of 4.04 heads per hectare. Thus, the reviewer's statement that Parcel A was primarily grazed by cattle in 2014 may be misleading.*

*We are aware of the 3% assumption and while this approach could be taken, we were not able to follow the numbers presented by the reviewer. Possibly some additional explanation could be provided on how the values given were derived.*

*At the same time, we propose another approximation for methane emissions from enteric fermentation from cattle as follows and in relation to the study by Felber et al. (2015). Felber et al. reported an average of 404 g CH4 per head per day in a table summarizing different. Taking this value and given the cattle occupied Parcel A (2.2 ha) for about four weeks with an average stocking rate of 4.04 heads per hectare (average of 12.5 and 5.3 for 2.2 hectares) our calculations are as follows.*

*Emissions for enteric methane = 404 g CH4/head/day * 4.04 head/ha * 30 days / 1000 to derive kg)*

*The total CH4 emissions calculated are thus 48.96 kg CH4 per ha. When we convert this to C, we derive emissions of 4.07 kg CH4-C per ha. This would be the value we expect also to see with the EC flux tower under perfect conditions with a non-movable point source. Unfortunately, such perfect conditions aren't reality and we may not have captured all of these emissions due to shifts in wind direction,*

*changes in turbulence as well as the actual animal movement. Also, as indicated by Felber et al. (2015) distance from the cow to the EC tower determines how much methane one measures with the EC tower. Moreover, 4 kg CH4-C are of minimal influence for both the C budget as well as for the GHG budget of the site (see Table 4). The proposed way forward is to add this information on the issue of grazing in the results section, ie the calculation provided here in a first step and secondly highlight that methane remains of minor importance at this site for both the C as well as the GHG budget, even if we were adding another 4 kg CH4-C in the year 2014.*

*To clarify on how the grazing removal was estimated and dealt with in the budget please see the following explanation. Grazing removal was quantified experimentally by having areas in both parcels from which the animals were excluded. At the end of each grazing period, the grass in the enclosures was cut similar to the approach taken when estimating harvests with subsequent laboratory analysis for C and N. Grazing is included in the harvest in Table 4, as this is a removal of biomass from the system. We agree that we had not included the return of nutrients via excreta (approx. 32% C, (Felber et al., 2016)) and will include this in the revised budget calculations for both C and N. This adjustment does not change the key results of the paper presented.*

Moderate Concerns
3. The focus (or perhaps title?) of this manuscript needs sharpening. The title indicates a focus on pasture restoration which is matched by the abstract, yet much attention is given to methodological considerations. Specific goal (ii) states "briefly compare two different measurement techniques" however the first two-thirds of the discussion (i.e. not briefly) comments on this aspect! While important and noteworthy, either change the title/abstract, or return the primary focus of the discussion to management effects. Additionally, goal (iii) is not really explored in this manuscript – perhaps combine with goal (i)?

*Thank you for this suggestion. In the revised manuscript we will combine goals (i) and (iii) and suggest shortening the discussion on the methodological aspects and give with this more attention to the primary goal of the study.*

4. Providing a partial N budget provides little useful information. Including individual components is beneficial, but to sum them up as an incomplete "budget" is not. If the authors choose to retain the N budget, please include some further context including some ballpark estimates of the remaining components to aid interpretation.

*Agreed. Since we intend to keep the N budget, we will add information by including ballpark estimates on ie. ammonia emissions, N deposition etc. in the revised manuscript. We will also include the necessary information on the origin of these estimates in the revised methodology section.*

5. While N2O flux gap filling is difficult, the use of running medians may be problematic, and especially for gaps occurring during pulse emissions (e.g. the restoration period/fertiliser applications). The authors should comment on limitations of this approach, especially in the absence of any uncertainties (which I accept is rarely done in N2O flux studies so do not see them as a requirement here).

*This is a very relevant point made by the reviewer. The method chosen here, follows the approach taken by Hoertnagl et al. (2018) and whom identified the running median being the most appropriate method to use if either large amounts of original data are available (ie as provided by the EC method) and/or if it is likely that the majority of N2O pulses have been covered by ie chamber measurements. Certainly, there are other options to fill N2O flux measurements and these were highlighted for instance in Nemitz et al. (2019) or Mishurov and Kiely (2011). Particularly, Nemitz et al. (2019) suggests linear interpolation for short gaps and daily averages to fill other gaps. For very long gaps more sophisticated and complex approaches such as machine learning tools are suggested.*
*Given that we aimed at deriving an annual budget which is relatively conservative we chose the running median approach. First of all, this way we are less likely to overestimate N2O emissions compared to ie the daily average approach. Linear interpolation would also have led to an overestimation of N2O*

*emissions particularly for the years 2010 and 2011 with few data points. Certainly, we see the lowest influence of gap filling errors for the years with EC measurements, whereas there may be a larger bias for the year with chamber measurements. Based on our 5-year observation period that indicated N2O emissions peaks during the growing season only and following fertilization events primarily (except 2012), we are confident that we covered the majority of these peaks during the years 2010 and 2011 when only chamber measurements are available. Nevertheless, and in line with comments made by reviewer #1 as well as our response to these comments during the access review phase, we intend to include additional gap-filling method estimates in the revised version of the manuscript as supplementary information.*

Minor/Technical Concerns
Lines 33-34: grazing is listed as both a regular and sporadic management activity. Please clarify which it is.

*We apologize for the mislead in wording and will rephrase as follows: "Grazing is a typical management activity in such intensive grassland. At our site, we observe grazing with either sheep or cattle for few days at the beginning or end of most years."*

Line 37: Missing the word "out" (or similar) after "carried".

*Done*

Lines 86-89: Why did you hypothesis continuous losses of CO2? Several studies (e.g. (Rutledge et al., 2017; Ammann et al., 2020, etc) show CO2 uptake in restoration and later years.

*Thank you for pointing this out. Actually, we had the hypothesis of increased CO2 uptake already in the manuscript (L. 89-90). We reworded these lines as follows: Prior to our measurements we hypothesized short-term losses of $CO_2$ after restoration and more continuous losses of primarily $N_2O$ following dramatic managements events such as ploughing occurring at irregular time intervals. We further hypothesized an increased carbon uptake strength compared to the pre-ploughing years.*

Lines 89-90: If you expect CO2 losses (as per the above point), why would you expect a C gain? Please adjust this and align with the previous sentence to clarify your hypothesis.

*See our comment to the previous remark made by the reviewer.*

Line 108: Do you mean CH4 emissions from the land or the grazers? In fact, this point needs clarity throughout the manuscript – are the grazers included within the system boundary, and therefore their emissions?

*We actually refer to both, land emissions/uptake as well as CH4 emissions from grazers. In terms of system boundaries, these are set to the ecosystem here, thus we account for the GHG emissions made by grazers (CH4 from enteric fermentation, as well as CH4 and N2O from excreta). Given that stocking rate was low and the actual time of grazing short we expected little effects of grazing on the budget while still aiming at being inclusive as we wanted to include all the management activities occurring in this field with some having a clear influence on GHG flux measurement, while others may not. We further included the offtake due to grazing in the budget calculations and revise the existing budgets by accounting for the returns of nutrient to the pasture via excreta deposition.*

Lines 123-127: this sentence is very clunky – suggest reviewing.

*We are not sure what the reviewer refers to here as these are two sentences in the original manuscript. However, in order to increase the flow of reading the suggested lines will be adjusted as follows in the revised manuscript. "The study by Hörtnagl et al. (2018) further elaborated the variation in management intensity and related variations in GHG exchange across sites, stressing the need for more*

*case studies based on continuous GHG observations to improve existing knowledge and close remaining knowledge gaps. To complete the picture on factors impacting ecosystem GHG exchange, irregular occurring events such as dry spells or extraordinary wet periods can further lead to enhanced or reduced GHG emissions (Chen et al., 2016; Hartmann and Niklaus, 2012; Hopkins and Del Prado, 2007; Mudge et al., 2011; Wolf et al., 2013)"*

Line 130: "adaptations" should be "adaptation" (no "s").

*Done*

Line 137: "respectively" is not needed – please delete.

*Done*

Lines 232-234: If an LI-7500 (rather than LI-7500A) was the self-heating correction applied?

*That was an oversight and we added the A.*

Lines 241-249: It was unclear to me what QA/QC procedures were applied to the raw (10/20Hz) and which to the 30-minute data. I suggest improving the clarity here.

*We rephrased this section by clearly distinguishing between raw data and raw time series (high frequency) and specifically state when we refer to 30-minute data.*

Line 248: what was considered the physically plausible range? Please include this information.

*Done*

Line 280: Order of words: "no longer closed" should be "closed no longer".

*Done*

Line 314: Remove the word "Up"

*Done*

Line 413: Insert the word "and" between "(Figure 1c)" and "temperatures".

*Done*

Lines 477-478: I think the before and after restoration periods should be separated. I don't believe averaging the two periods to be fair as part of the purpose of restoration is to improve growth, and therefore modification of CO2 exchange should also be expected.

*This may be a misunderstanding. We clearly differentiate between periods as indicated in the original mansuscript under sections 3.3. CO2 exchange and N2O exchange as well as under section 3.4.*

Line 480: According to Table 2, CH4 emissions for 2013 and 2014 were actually >1 – please correct.

*This is correct for the years 2012, 2013 and 2014 and the values seen are very similar to values reportend by Felber et al. (2015). Given the magnitude of the other GHG fluxes, methane remains a minor contribution to both the C as well as the GWP budgets (less than half the contribution of N2O for the years 2013 and 2014 which are dominated by the CO2 signal).*

Line 538: Correct the format of the reference

*Done*

Line 579-580: Are you referring to the measured CO2 exchange to be _50 g C m-2 y-1, or the uncertainty? This sentence is very unclear as no uncertainty has been presented, so please clarify.

*This refers to the statement made by Baldocchi et al. 2003, whom stated that annual numbers presented from EC measurements can vary by as much as by 50 g C per year. Thus, we want to encourage that this is an uncertainty anyone should keep in mind when evaluating annual budgets derived by the EC technique.*

Table 1: I find the "max data availability" columns repetitive – perhaps just a single column of this data?

*Good point, thank you! We will remove the repetitive statement of numbers in the revised manuscript.*

Table 4: I suspect the labelling of Parcels A and B for both fertilizer and harvest are not correct. As written, fertilizer was only applied to Parcel A, and Harvest to Parcel B. Please correct is appropriate.

*This is actually only an incorrect labelling and should refer to harvest for Parcel A and B as well as fertilizer for Parcel A and B. This will be corrected in the revised manuscript.*

References
Ammann, C., Neftel, A., Jocher, M., Fuhrer, J., Leifeld, J., 2020. Effect of management and weather variations on the greenhouse gas budget of two grasslands during a 10- year experiment. Agric. Ecosyst. Environ. 292.
Felber, R., Bretscher, D., Münger, A., Neftel, A., Ammann, C., 2016. Determination of the carbon budget of a pasture: effect of system boundaries and flux uncertainties. Biogeosciences 13, 2959-2969.
Rutledge, S., Wall, A.M., Mudge, P.L., Troughton, B., Campbell, D.I., Pronger, J., Joshi, C., Schipper, L.A., 2017. The carbon balance of temperate grasslands part II: The impact of pasture renewal via direct drilling. Agric. Ecosyst. Environ. 239, 132-142.

*We thank the reviewer for pointing us towards these references and we refer to these in the revised version of the manuscript.*

---

## Author Comment (AC2) · 19 Aug 2020

We would like to thank Reviewer #1 once again for the insights provided during the access review and repost our original response on how we foresee to revise the current manuscript (during the access review stage the reviewer was referred to as reviewer #2). We would like to further point the reviewer to the response given to the full review provided by Reviewer #2 (Reviewer #1 in the Access Review) whom had similar concerns concerning the gap-filling of N2O flux measurements.

Please also note the supplement to this comment:

[Figure]

https://bg.copernicus.org/preprints/bg-2020-141/bg-2020-141-AC2-supplement.pdf

[Figure]

**Supplement:**

**Response to Access Reviews:**

Reviewer #1:
We would like to thank reviewer #1 for the positive evaluation, the identification of the small glitches in the text and thus implemented all the suggest corrections. Our responses are given in italic font.

Technical corrections requested by Reviewer #1:

Typing corrections:
Line 30: year "2014" is written as "214" - *DONE*
Line 47: "as well" should be "as well as" - *DONE*
Line 48: "… during all" seems an incomplete sentence – *DONE (we added "years")*
Line 132: "ecosystem" should be "ecosystems" - *DONE*
Line 202: Fuchs et al reference states the years as "2081" – presumably this should be "2018" (also use of square brackets?) - *DONE*
Line 345: Should "bon" be "box"? - *DONE*
Table 1: Max data availability for 2010, 2011, 2013 & 2014 should be 17520 not 17521
Table 4: Please differentiate Parcel A and B for Fertilizer and Harvest (both currently labelled as Parcel A) - *DONE*
Figure 4 caption: please remove "(a)" immediately after "Figure 4:" - *DONE*

Figure and Table Clarifications:
All applicable tables and figures: please indicate sign convention either in the caption or on the figure (e.g. Figure 2, that positive values represent and export) - *DONE*
Figure 3: Please indicate what the grey lines represent (presumably uncertainty or standard deviation?) - *DONE*
Figure 4a: Please state in the caption what the grey lines represent (half-hours fluxes?) - *DONE*

Reviewer #2:

We would like to thank reviewer #2 for the overall positive evaluation and for providing feedback on the points that the reviewer encourages to be addressed. For this access review, we implemented all technical corrections right away and lay out a way forward on how we intend to address the currently raised points following the end of the full review period in a fully revised version of the manuscript. Our responses are given in italic font.

The manuscript "Memory effects on greenhouse gas emissions (CO2, N2O and CH4) following grassland restoration?" by Merbold et al. is a well written long term study of GHGs from a grazed grassland system in Switzerland. The team have used a mixture of measurement methods over a 5 year period to get a very good picture of a full GHG budget for the field. This is a very valuable study as such long term observations are rare and it answers some questions that are not well studied.
I found the manuscript interesting to read, and it was written to a very high standard and I do believe that it should be published after some amendments.
I do have some comments that I feel should be addressed by the authors that I believe would improve the quality and usefulness of the study for others. Although these comments

are numerous and not entirely simple to address, if the authors can amend their study to incorporate them I feel the work would benefit greatly.

*Thank you for the positive evaluation and we suggest ways forward point by point below.*

1. A large assumption made by the study is that the eddy covariance measurements are entirely truthful of the conditions in the field. It has been observed in the past that long-term carbon budgets derived from eddy covariance can be biased due to assumptions made by the method. Often negative carbon fluxes are reported in similar systems, however, when investigating deep soil cores there was found to be no significant difference in C content of the soil (see Jones et al., doi:10.5194/bg-14-2069-2017 for one such study). The manuscript does not provide evidence of the C stock in the soil beyond the Eddy C measurements to back up the evidence which would have made it a much more significant study. This does not invalidate the study by any means, but without clarification of potential uncertainties, it increases the danger that the study provides "concrete" evidence of mitigation methods (i.e. grazing animals is a carbon sink) that has been used recently by advocates of the meat industry to justify the long-term environmental aspects of livestock farming. I would advise a short message of discussion to highlight that there is room for error in the measurements and that soil carbon was not measured to validate the measurements. Alternatively, if the soil measurements are there, please include them.

*Indeed, we agree that soil inventories should be linked to EC fluxes more often, particularly since EC measurements are often seen as entirely truthful. We are confident that the EC method is a valuable and powerful tool to investigate C and N fluxes at ecosystem scale – not necessarily the exact entire field as suggested in the literature ( Hill et al. 2016* https://doi.org/10.1111/gcb.13547) *– yet, it allows to derive a general view on whether a system is likely to gain/loose carbon/nitrogen. We further agree that continuous flux measurements and thus budgets should be validated with other independent methods, ie a soil inventory. Yet, determining changes in soil C/N is similarly not trivial and takes considerable time as suggested by ie. Schrumpf et al. 2011* https://doi.org/10.5194/bg-8-1193-2011*. Additional, within this specific project we were not able to carry out a resampling of* the soils while further advocating for this in follow-up projects. Certainly, we will aim at implementing a thorough uncertainty assessment of the numbers presented in the revised version of the manuscript. Multiple approaches to estimate the uncertainty in EC flux measurements as well as in gap-filling methods are available (ie Post et al. 2015 https://doi.org/10.5194/bg-12-1205-2015, Vitale et al 2019 https://doi.org/10.1007/s00477-019-01664-4, Hollinger and Richardson 2005 https://doi.org/10.1093/treephys/25.7.873, Nicolini et al. 2018, https://doi.org/10.1016/j.agrformet.2017.09.025)

2. I do not agree with the way that the N2O flux data has been handled in the study. N2O fluxes measured using chambers almost always follow a log-normal distribution in space, so any data analysis must take this into account when handling means and uncertainties. A simple arithmetic mean with associated uncertainty (not sure what the error bars on Fig 3 and 4 represent?) will not be an adequate way to represent

this data (although commonly used wrongly in previous studies). This will result in a skewing of the data and large overestimates in minimum confidence intervals and underestimations of maximum confidence intervals. An example is when uncertainties of N2O cross the negative threshold when no observations of flux dip below zero. This is not a satisfactory way to present the data. I recommend using a more sophisticated analysis technique and showing 95% confidence intervals where possible for a thorough comparison of the measurement techniques.

*We thank the reviewer for the critical assessment. Our approach followed the method used by Hoertnagl et al. 2018* https://doi.org/10.1111/gcb.14079. *We already added here – as technical correction - the information on the grey lines (standard deviation). Furthermore, and following the full review process we anticipate to take the log-normal distribution into account.*

3. L303: Due to the log-normal distribution of N2O emissions measured using chambers, most measurements will be very close to zero and ppb differences in gas samples will hover around detection limits of the analysis instrument. In such cases, the R2 value of the fits will be very low for many, but the regression between points will still be valid (effectively an average of the instrument noise with a slope near zero). By cutting data with R2 lower than 0.8 I assume that a very large number of small fluxes are removed from the dataset. If this is the case I would recommend a threshold on this QC method, or a more detailed explanation of what impact this had on the data in the text if this is not the case (as I read it, the method would likely contribute to a large bias is flux estimates).

*We implemented thorough QC criteria concerning the N2O flux calculations. All the details have been in detail provided in Imer et al. 2013* https://doi.org/10.5194/bg-10-5931-2013 *stating the r2 threshold of 0.8. Overall, the low fluxes being part of our observations were not being the limit of detection and have thus been included in this study. In the revised manuscript we will further clarify out QA/QC approaches in order to avoid misinterpretation.*

4. Uncertainties in cumulative emissions are not presented which makes it difficult to compare with other studies or what impact gap-filling and weather may have had on the study. This should be easily manageable for CO2 for which models exist, and probably for CH4 using simple gap-filling as it was found to be approximate zero throughout the study. I understand that there is no definitive way to gap-fill N2O, however a running median is not a statistically defensible way to "model" data. As a result no uncertainty will be calculated from this method. If the authors want to estimate uncertainties in cumulative N2O fluxes, they will have to develop a more sophisticated approach to gap-filling.

*We agree with the reviewer and will implement such uncertainty estimates, in particular for CO2 and CH4, in the revised version of the manuscript following the full review process. The Running median approach was chosen, following Hoertnagl et al. 2018 (see above) as this seemed at the time being the best possible way to fill N2O flux data gaps. In terms of*

*uncertainty, we intend to present additional gap-filling methods for N2O in the revised manuscript.*

5.  I feel a nitrogen budget without NH3, NOx and N2 is not very useful. Combined, these gases will likely contribute approx. 50% of nitrogen losses from the system. Perhaps a better way to confer N losses is to calculate the emission factors of the fertiliser applications, as that is a more generally used term for such activities in literature and is a better description of the presented results in the study.

*We are in full agreement with the reviewer that other N compounds build a large part of the N budget. In order to proceed we suggest two points: (1) We will further clarify that we are only showing a partial N budget caused by the fact that we do not have data available for NH3, NOx and N2 losses, and (2) to incorporate emission factors based on the losses observed via N2O and the fertilizer inputs in the revised manuscript.*

6.  Is there a way to estimate the N content of the fodder/grass on the field before tillage to assess the emissions from the herbage being tilled into the soil?

We have thought about this too when preparing the manuscript and must confess that we had not taken such measurements. However, to our current knowledge the additional N being incorporated into soil during tillage should be small caused by relatively low vegetation cover/height at this time of the year. For the revised manuscript, we will provide a better estimate based on existing literature.

7.  Does the carbon budget take into account vehicle use? Is it insignificant or does tractor diesel have a role to play?

*The currently presented budget does not include C emissions from vehicle use for two reasons: (1) the hours farm vehicles are being used on this field are very limited over the course of the year given the small size of the fields (negligible). The negligibility of these emissions was further underlined (2) by a MSc thesis that investigated full farmgate budgets in the years prior this study. We will add an additional sentence concerning this point in the revised version of the manuscript.*

8.  L225: Can you explain what you mean by an internal reference cell in the instrument for the QCLAS? To my knowledge, these cells are used to find absorption lines on the spectra and not for calibration as they leak over time. The QCLAS system typically does not require calibration as it operates on the principles that the absorption follows Hitran quantum mechanics laws.

*Thank you for this comment and this seems to be a misunderstanding of what we have written. We stated that the infrared gas analyser was calibrated regularly, while we also wrote that the QCLAS was fitted against an internal reference cell. In order to create better clarity we changed this sentence as follows: "The QCLAS did not need calibration due to its*

*operating principles, and an internal reference cell (mini-QCL manual, Aerodyne Research Inc., Billerica, MA, USA) eased finding the absorption spectra after each restart of the analyzer."*

Some minor corrections
L283: I think there is a bit of wording here that is confusing. Flushing the chamber with the syringe isn't technically correct. I think it would be better to say that the syringe was used to pump the chamber to circulate the air to avoid the concentration gradients? - *DONE*
L471: here the order of the sentences makes it sound like CH4 contributed to 70% of the budget. Please re-order. – *DONE, we added "*the contribution of $CO_2$*"*
L606: Change highlight to highlights - *DONE*

---

## Author Response (AR1)

**Response to reviewers on our manuscript "Memory effects on greenhouse gas emissions (CO2, N2O and CH4) following grassland restoration?" by Lutz Merbold et al.**

We thank both reviewers for their critical assessment and provide a revised manuscript addressing the reviewer's comments. Throughout the following document, the reviewer's comment is stated first, followed by our response in *italic* font. We further attach a clean revised manuscript and a track changed version for ease of the editor and reviewers.

Reviewer #1:
The study presented here title "Memory effects on greenhouse gas emissions (CO2, N2O and CH4) following grassland renovation?" presents trace gas measurements from 5 years of a grazed and harvest pasture in Switzerland including a pasture restoration event. In general, this is a well written and worthwhile study. Few studies report all greenhouse gases, and even fewer for multiple years and covering infrequent management activities. I believe this to be of publication quality following consideration of my commentary below. I have separated my comments into major, moderate and minor/technical concerns based on importance and impact to the manuscript as I see it. I believe these can be dealt with by the authors and would further enhance the manuscript.

*We thank the reviewer for this positive assessment and share the opinion of few studies reporting on multiple years of GHG exchange measurements of the three GHGs covering specific management activities.*

Major concerns
1. CH4 fluxes: I have major concerns with the usage of the CH4 fluxes as presented in this manuscript. Firstly, while the authors present a comparison of N2O chamber and eddy covariance data (Figure 3), they do not for CH4. I believe this is likely as the comparison does not suggest any 1:1 relationship (based on my interpretation of Figure 4b). The authors then use this chamber data to derive annual CH4 fluxes for the years without EC data and assume to be comparable with the EC derived annual fluxes. From the data presented, I see no evidence to believe this to be the case (unlike N2O). Given the two chamber years suggest a small uptake of CH4, while the last three a release of CH4 coinciding with a difference in measurement methodology, I question whether the authors really believe these years are comparable. While the authors discuss these methodology differences in detail in the discussion section, and overall the contribution of CH4 to the GHG budget is small, I believe further attention needs to be given to this, and ideally the equivalent plot to figure 3b is presented for CH4. Based on the timing of management events (pasture restoration) and change in measurement methodology it could be easily interpreted as pasture restoration changes grassland CH4 exchange from an uptake to release.

*These are indeed relevant points and surely, we do not want to give the impression that pasture restoration changes grassland CH4 exchange from an uptake to release as this can not be proven by the data presented in this study (see following response). We had preferred to show a similar comparison as given for N2O, however the methane concentrations measurements were not reliable in 2013 due to a flame ionization detector (FID) malfunction in the gas chromatograph.*
*Overall, we also did not expect to find a similar relation between the methane flux measurements obtained by eddy covariance and chambers caused by the small magnitude of the fluxes measured. As stated in the original manuscript "We calculated detection limits for the individual GHGs from our manual chambers following (Parkin et al., 2012). Detection limits were $0.34 \pm 0.26$ nmol m$^{-2}$ s$^{-1}$, $0.05 \pm 0.02$ nmol m$^{-2}$ s$^{-1}$, and $0.06 \pm 0.06$ µmol m$^{-2}$ s$^{-1}$ for CH$_4$, N$_2$O and CO$_2$, respectively, clearly indicating that methane fluxes measured by GHG chambers in 2010/2011 were on average $-0.16 \pm 0.16$ nmol CH$_4$ m$^{-2}$ s$^{-1}$, (see Table 2) and thus below the actual detection limit."*
*However, we did compare our eddy covariance methane flux values (methane fluxes fluctuating around 0 with an overall range of -40 up to +40 nmol CH4 m-2 s-1 (Figure 4 b)) with the values reported by* (Felber et al., 2015) *from a similar grassland system in Western Switzerland.* (Felber et al., 2015) *have shown that such values measured by the EC technique represent a soil signal (Figure 6 in* Felber et al. 2015*).*

*Following this, we agree that we should not have computed annual sums for the years 2010/2011 for methane and will remove these in the revised manuscript. We will only present the gap-filled numbers for methane for 2012 -2014 and show the actual measurements derived with GHG flux chambers for the years 2010/2011 only (Figure 4b).*

*Overall, we would like to point out again that methane fluxes are of minor importance for the carbon and greenhouse gas budget of the site under the current management (see also our response to the second concern as well as the concern made by reviewer #2 on the influence of grazing animals on methane fluxes).*

2. The impact of grazing needs further consideration. While harvesting is more common in this study, the impact of grazing needs further clarification and/or modification of the presented results. Firstly, it is unclear to me how the grazing off-take was estimated (please clarify), and whether the deposition of excreta C was included in the C balances. While I'm not familiar with sheep grazing, at least for cattle this can be in the order of one-third of consumption, and therefore not an insignificant component (especially for 2014, Parcel A with 1769.9 kg C ha-1 of grazing removal according to table S1) and requiring acknowledgement of how this is currently dealt with, or included in the C balance (e.g. Table 2).

Furthermore, the authors state they did not detect any CH4 release with grazing (lines 432-433). Using the example of Parcel A in 2014, which was primarily grazed by cattle, and assuming _3% was converted and released as CH4 (e.g. Felber et al. (2016)), 53.1 kg C ha-1 would have been emitted from the grazers as CH4, which when converted to g CO2-eq m-2 calculated to 240 g CO2-eq m-2 or much larger than the 55 g CO2-eq m-2 reported in table. If this was not detected, then I suggest the authors reconsider how grazing related CH4 is dealt with in this manuscript given they are reporting ecosystem scale GHG budgets.

*Indeed, methane emissions from grazing animals need to be considered in annual budgets of methane and carbon. We argue that these are already accounted for in our data, since our observation boundary is the ecosystem and thus, we only include CH4 from animals when these are on the field. Grazing intensity was extremely low and only lasted for few days in the specific years (2010, 2011, 2014). Also, most of the grazing were sheep, and cattle were only present in 2014 in Parcel A for less than four weeks in total at an average stocking rate of 4.04 heads per hectare. Thus, the reviewer's statement that Parcel A was primarily grazed by cattle in 2014 is incorrect.*

*Furthermore, we are aware of the 3% assumption and while this approach could be taken, we were not able to follow the numbers presented by the reviewer. Possibly some additional explanation could be provided on how the values given were derived.*

*At the same time, we propose another approximation for methane emissions from enteric fermentation from cattle as follows and in relation to the study by* Felber et al. (2016). Felber et al. (2016) *reported an average of 404 g CH4 per head per day in a table summarizing several studies. Taking this average value and given the cattle occupied Parcel A (2.2 ha) for about four weeks with an average stocking rate of 4.04 heads per hectare (average of 12.5 and 5.3 for 2.2 hectares) our calculations are as follows.*

*Emissions for enteric methane = 404 g CH4/head/day * 4.04 head/ha * 30 days / 1000 to derive kg)*

*The total CH4 emissions calculated are thus 48.96 kg CH4 per ha (4.89 g CH4 m-2). When we convert this to C, we derive emissions of 4.07 kg CH4-C per ha (0.40 g CH4-C m-2). This would be the value we expect also to see with the EC flux tower under perfect conditions with a non-movable point source. Unfortunately, such perfect conditions are not the reality and we may not have captured all of these emissions due to shifts in wind direction, changes in turbulence as well as the actual animal movement out of the fetch. Also, as indicated by Felber et al. (2016) distance from the cow to the EC tower determines how much methane one measures with the EC tower. Moreover, 4.07 kg CH4-C ha-1 (0.40 g CH4-C m-2) are of minimal influence for both the C budget as well as for the GHG budget of the site (see Table 4). In order to clarify this point, we add this information on the issue of grazing in the revised manuscript.*

*Grazing removal was quantified experimentally by having areas in both parcels from which the animals were excluded. At the end of each grazing period, the grass in the enclosures was cut similar to the approach taken when estimating harvests with subsequent laboratory analysis for C and N. Grazing is included in the harvest in Table 4, as this is a removal of biomass from the system. The return of nutrients via excreta ((approx. 32% C, (Felber et al., 2016)) resembles a recycling of nutrients within the systems and associated GHG emissions would be included in the EC measurements. Following our previous argument of the very low stocking density, this is unlikely to have considerable effects to the results of the study.*

Moderate Concerns

3. The focus (or perhaps title?) of this manuscript needs sharpening. The title indicates a focus on pasture restoration which is matched by the abstract, yet much attention is given to methodological considerations. Specific goal (ii) states "briefly compare two different measurement techniques" however the first two-thirds of the discussion (i.e. not briefly) comments on this aspect! While important and noteworthy, either change the title/abstract, or return the primary focus of the discussion to management effects. Additionally, goal (iii) is not really explored in this manuscript – perhaps combine with goal (i)?

*Thank you for this suggestion. In the revised manuscript we combined the goals (i) and (iii) and shortened the discussion on the methodological aspects while giving more attention to the primary goal of the study. As a consequence, the former objective (iv) has now become the new objective (iii) (see the version of the manuscript with track changes).*

4. Providing a partial N budget provides little useful information. Including individual components is beneficial, but to sum them up as an incomplete "budget" is not. If the authors choose to retain the N budget, please include some further context including some ballpark estimates of the remaining components to aid interpretation.

*We agree that particularly in terms of N providing the partial budget is not as good as providing a full N budget. At the same time, we avoided after careful consideration, to provide a N budget with ballmark estimates as some fluxes would be largely uncertain due to little data availability from such systems (ie nitrate leaching) or overall limited data availability across agricultural systems (ie losses in form of NOx and N2). Yet we are aware that losses of nitrogen via ie NH3, N2, NOx can be much larger than the losses via N2O. Consequently, we rephrased the respective objective (previously iv now iii) to "(iii) to provide a GHG budget of the site". We further changed the wording from C and N budgets to C and N gains and losses with the losses we specifically refer to losses of N via $N_2O$.*

5. While N2O flux gap filling is difficult, the use of running medians may be problematic, and especially for gaps occurring during pulse emissions (e.g. the restoration period/fertiliser applications). The authors should comment on limitations of this approach, especially in the absence of any uncertainties (which I accept is rarely done in N2O flux studies so do not see them as a requirement here).

*This is a very relevant point made by the reviewer. The method chosen here, follows the approach taken by Hoertnagl et al. (2018), whom identified the running median being the most appropriate method to use if either large amounts of original data are available (ie as provided by the EC method) and/or if it is likely that the majority of N2O pulses have been covered by ie chamber measurements. Certainly, there are other options to fill N2O flux measurements and these were highlighted for instance in Nemitz et al. (2019) or Mishurov and Kiely (2011). Particularly, Nemitz et al. (2019) suggests linear interpolation for short gaps and daily averages to fill other gaps. For very long gaps more sophisticated and complex approaches such as machine learning tools are suggested.*
*Given that we aimed at deriving an annual budget which is relatively conservative we chose the running median approach. First of all, this way we are less likely to overestimate N2O emissions compared to ie the daily average approach. Linear interpolation would also have led to an overestimation of N2O*

*emissions particularly for the years 2010 and 2011 with few data points. Certainly, we see the lowest influence of gap filling errors for the years with EC measurements, whereas there may be a larger bias for the year with chamber measurements. Based on our 5-year observation period that indicated N2O emissions peaks during the growing season only and following fertilization events primarily (except 2012), we are confident that we covered the majority of these peaks during the years 2010 and 2011 when only chamber measurements were available. Thus, we decided to remain with the chosen approach as we do not think it is beneficial to state values which are likely to be more biased than the chosen approach.*

Minor/Technical Concerns
Lines 33-34: grazing is listed as both a regular and sporadic management activity. Please clarify which it is.

*We apologize for the mislead in wording and will rephrase as follows: "Grazing is a typical management activity in such intensive grassland. At our site, we observe grazing with either sheep or cattle for few days at the beginning or end of most years."*

Line 37: Missing the word "out" (or similar) after "carried".

*Done*

Lines 86-89: Why did you hypothesis continuous losses of CO2? Several studies (e.g. (Rutledge et al., 2017; Ammann et al., 2020, etc) show CO2 uptake in restoration and later years.

*Thank you for pointing this out. Actually, we had the hypothesis of increased CO2 uptake already in the manuscript (L. 89-90). We reworded these lines as follows: Prior to our measurements we hypothesized short-term losses of $CO_2$ after restoration and more continuous losses of primarily $N_2O$ following dramatic managements events such as ploughing occurring at irregular time intervals. We further hypothesized an increased carbon uptake strength compared to the pre-ploughing years.*

Lines 89-90: If you expect CO2 losses (as per the above point), why would you expect a C gain? Please adjust this and align with the previous sentence to clarify your hypothesis.

*See our comment to the previous remark made by the reviewer.*

Line 108: Do you mean CH4 emissions from the land or the grazers? In fact, this point needs clarity throughout the manuscript – are the grazers included within the system boundary, and therefore their emissions?

*We actually refer to both, land emissions/uptake as well as CH4 emissions from grazers. In terms of system boundaries, these are set to the ecosystem here, thus we account for the GHG emissions made by grazers (CH4 from enteric fermentation, as well as CH4 and N2O from excreta) for the years 2012-2014. Given that stocking rate was low and the actual time of grazing short we expected little effects of grazing on the budget while still aiming at being inclusive as we wanted to include all the management activities occurring in this field. We further included the offtake due to grazing in the budget calculations. The recycling of nutrients from grazing animals and their deposits is included in the eddy covariance measurements. While this may not be the case for 2011/2012. Given the small stocking rate and as explained before this is likely of minor importance and surely will not change the results.*

Lines 123-127: this sentence is very clunky – suggest reviewing.

*We are not sure what the reviewer refers to here as these are two sentences in the original manuscript. However, in order to increase the flow of reading the suggested lines will be adjusted as follows in the revised manuscript. "The study by Hörtnagl et al. (2018) further elaborated the variability in management intensity and related variations in GHG exchange across sites, stressing the need for more*

*case studies based on continuous GHG observations to improve existing knowledge and close remaining knowledge gaps. To complete the picture on factors impacting ecosystem GHG exchange, irregular occurring events such as dry spells or extraordinary wet periods can further lead to enhanced or reduced GHG emissions (Chen et al., 2016; Hartmann and Niklaus, 2012; Hopkins and Del Prado, 2007; Mudge et al., 2011; Wolf et al., 2013)"*

Line 130: "adaptations" should be "adaptation" (no "s").

*Done*

Line 137: "respectively" is not needed – please delete.

*Done*

Lines 232-234: If an LI-7500 (rather than LI-7500A) was the self-heating correction applied?

*That was an oversight and we added the A. The correction was applied.*

Lines 241-249: It was unclear to me what QA/QC procedures were applied to the raw (10/20Hz) and which to the 30-minute data. I suggest improving the clarity here.

*We rephrased this section by clearly distinguishing between raw data and raw time series (high frequency) and specifically state when we refer to 30-minute data.*

Line 248: what was considered the physically plausible range? Please include this information.

*Done*

Line 280: Order of words: "no longer closed" should be "closed no longer".

*Done*

Line 314: Remove the word "Up"

*Done*

Line 413: Insert the word "and" between "(Figure 1c)" and "temperatures".

*Done*

Lines 477-478: I think the before and after restoration periods should be separated. I don't believe averaging the two periods to be fair as part of the purpose of restoration is to improve growth, and therefore modification of CO2 exchange should also be expected.

*This may be a misunderstanding. We clearly differentiate between periods as indicated in the original manuscript under sections 3.3. CO2 exchange and N2O exchange as well as under section 3.4.*

Line 480: According to Table 2, CH4 emissions for 2013 and 2014 were actually >1 – please correct.

*This is correct for the years 2012, 2013 and 2014 and the values seen are very similar to values reported by* Felber et al. (2015). *Given the magnitude of the other GHG fluxes, methane remains a minor contribution to the GWP budgets.*

Line 538: Correct the format of the reference

*Done*

Line 579-580: Are you referring to the measured CO2 exchange to be +/- 50 g C m-2 y-1, or the uncertainty? This sentence is very unclear as no uncertainty has been presented, so please clarify.

*This refers to the statement made by Baldocchi et al. (2003), who stated that annual numbers presented from EC measurements can vary by as much as +/- 50 g C m-2 y-1. Thus, we want to encourage that this is an uncertainty anyone should keep in mind when evaluating annual budgets derived by the EC technique.*

Table 1: I find the "max data availability" columns repetitive – perhaps just a single column of this data?

*Good point, thank you! We removed the repetitive statement of numbers in the revised manuscript and also removed the columns presenting the water fluxes as these are not referred to in the manuscript.*

Table 4: I suspect the labelling of Parcels A and B for both fertilizer and harvest are not correct. As written, fertilizer was only applied to Parcel A, and Harvest to Parcel B. Please correct is appropriate.

*This is actually only an incorrect labelling and should refer to harvest for Parcel A and B as well as fertilizer for Parcel A and B. This has been corrected.*

*We thank the reviewer for pointing us towards these references and we refer to these in the revised version of the manuscript.*

Reviewer #2:

We would like to thank reviewer #2 for the overall positive evaluation and for providing feedback on the points that the reviewer encourages to be addressed. Our responses to the questions/concerns are given in italic font.

The manuscript "Memory effects on greenhouse gas emissions (CO2, N2O and CH4) following grassland restoration?" by Merbold et al. is a well written longterm study of GHGs from a grazed grassland system in Switzerland. The team have used a mixture of measurement methods over a 5 year period to get a very good picture of a full GHG budget for the field. This is a very valuable study as such longterm observations are rare and it answers some questions that are not well studied.
I found the manuscript interesting to read, and it was written to a very high standard and I do believe that it should be published after some amendments.
I do have some comments that I feel should be addressed by the authors that I believe would improve the quality and usefulness of the study for others. Although these comments are numerous and not entirely simple to address, if the authors can amend their study to incorporate them I feel the work would benefit greatly.

*Thank you for the positive evaluation and we suggest ways forward point by point below.*

A large assumption made by the study is that the eddy covariance measurements are entirely truthful of the conditions in the field. It has been observed in the past that long-term carbon budgets derived from eddy covariance can be biased due to assumptions made by the method. Often negative carbon fluxes are reported in similar systems, however, when investigating deep soil cores there was found to be no significant difference in C content of the soil (see Jones et al., doi:10.5194/bg-14-2069-2017 for one such study). The manuscript does not provide evidence of the C stock in the soil beyond the Eddy C measurements to back up the evidence which would have made it a much more significant study. This does not invalidate the study by any means, but without clarification of potential uncertainties, it increases the danger that the study provides "concrete" evidence of mitigation methods (i.e. grazing animals is a carbon sink) that has been used recently by advocates of the meat industry to justify the long-term environmental aspects of livestock farming. I would advise a short message of discussion to highlight that there is room for error in the measurements and that soil carbon was not measured to validate the measurements. Alternatively, if the soil measurements are there, please include them.

*Indeed, we agree that soil inventories should be linked to EC fluxes more often, particularly since EC measurements are often seen as entirely truthful. We are confident that the EC method is a valuable and powerful tool to investigate C fluxes at ecosystem scale – not necessarily the exact entire field as suggested in the literature ( Hill et al. 2016 https://doi.org/10.1111/gcb.13547). Yet, it allows to derive a general view on whether an ecosystem is likely to gain/loose carbon. We further agree that continuous flux measurements and thus budgets should be validated with other independent methods, ie a soil inventory. Yet, determining changes in soil C/N is similarly not trivial and takes considerable time as suggested by ie. Schrumpf et al. 2011 https://doi.org/10.5194/bg-8-1193-2011. Additional, within this specific project we were not able to carry out a resampling of the soils while further advocating for this in follow-up projects. Multiple approaches to estimate the uncertainty in EC flux measurements as well as in gap-filling methods are available (ie Post et al. 2015 https://doi.org/10.5194/bg-12-1205-2015, Vitale et al 2019 https://doi.org/10.1007/s00477-019-01664-4, Hollinger and Richardson 2005 https://doi.org/10.1093/treephys/25.7.873, Nicolini et al. 2018, https://doi.org/10.1016/j.agrformet.2017.09.025) pointing towards the reliability of EC measurements. As we primarily provide a GHG budget – after having revised the objectives – these numbers do not represent a full farm-scale assessment.*

I do not agree with the way that the N2O flux data has been handled in the study. N2O fluxes measured using chambers almost always follow a log-normal distribution in space, so any data analysis must take this into account when handling means and uncertainties. A simple arithmetic mean with associated uncertainty (not sure what the error bars on Fig 3 and 4 represent?) will not be an adequate way to represent this data (although commonly used wrongly in previous studies). This will result in a skewing of the data and large overestimates in minimum confidence intervals and underestimations of maximum confidence intervals. An example is when uncertainties of N2O cross the negative threshold when no observations of flux dip below zero. This is not a satisfactory way to present the data. I recommend using a more sophisticated analysis technique and showing 95% confidence intervals where possible for a thorough comparison of the measurement techniques.

*We thank the reviewer for the critical assessment. Our approach followed the method used by Hoertnagl et al. 2018 https://doi.org/10.1111/gcb.14079. Hoertnagl et al. (2018), whom identified the running median being the most appropriate method to use if either large amounts of original data are available (ie as provided by the EC method) and/or if it is likely that the majority of N2O pulses have been covered by ie chamber measurements. Certainly, there are other options to fill N2O flux measurements and these were highlighted for instance in Nemitz et al. (2019) or Mishurov and Kiely (2011). Particularly, Nemitz et al. (2019) suggests linear interpolation for short gaps and daily averages to fill other gaps. For very long gaps more sophisticated and complex approaches such as machine learning tools are suggested.*

*Given that we aimed at deriving an annual budget which is relatively conservative we chose the running median approach. First of all, this way we are less likely to overestimate N2O emissions compared to ie the daily average approach. Linear interpolation would also have led to an overestimation of N2O emissions particularly for the years 2010 and 2011 with few data points. Certainly, we see the lowest influence of gap filling errors for the years with EC measurements, whereas there may be a larger bias for the year with chamber measurements. Based on our 5-year observation period that indicated N2O emissions peaks during the growing season only and following fertilization events primarily (except 2012), we are confident that we covered the majority of these peaks during the years 2010 and 2011 when only chamber measurements were available. Thus, we decided to remain with the chosen approach as we do not think it is beneficial to state values which are likely to be more biased than the chosen approach.*

L303: Due to the log-normal distribution of N2O emissions measured using chambers, most measurements will be very close to zero and ppb differences in gas samples will hover around detection limits of the analysis instrument. In such cases, the R2 value of the fits will be very low for many, but the regression between points will still be valid (effectively an average of the instrument noise with a slope near zero). By cutting data with R2 lower than 0.8 I assume that a very large number of small fluxes are removed from the dataset. If this is the case I would recommend a threshold on this QC method, or a more detailed explanation of what impact this had on the data in the text if this is not the case (as I read it, the method would likely contribute to a large bias is flux estimates).

*We implemented thorough QC criteria concerning the N2O flux calculations. All the details have been in detail provided in Imer et al. (2013), including the R2 threshold and how many data points were dismissed. Overall, the low fluxes being part of our observations were not being the limit of detection and have thus been included in this study.*

Uncertainties in cumulative emissions are not presented which makes it difficult to compare with other studies or what impact gap-filling and weather may have had on the study. This should be easily manageable for CO2 for which models exist, and probably for CH4 using simple gap-filling as it was found to be approximate zero throughout the study. I understand that there is no definitive way to gap-fill N2O, however a running median is not a statistically defensible way to "model" data. As a result no uncertainty will be calculated from this method. If the authors want to estimate uncertainties in cumulative N2O fluxes, they will have to develop a more sophisticated approach to gap-filling.

*We agree with the reviewer that there are different approaches to gapfill GHG flux data. Certainly, the gapfilling approaches for CO2 and CH4 are better developed than for N2O. The running median approach was chosen, following Hoertnagl et al. 2018 (see above) as this seemed at the time being the best possible way to fill N2O flux data gaps given the ecosystem observed.*

I feel a nitrogen budget without NH3, NOx and N2 is not very useful. Combined, these gases will likely contribute approx. 50% of nitrogen losses from the system. Perhaps a better way to confer N losses is to calculate the emission factors of the fertiliser applications, as that is a more generally used term for such activities in literature and is a better description of the presented results in the study.

*We are in full agreement with the reviewer that other N compounds build a large part of the N budget. We thus adjusted the manuscript to only show the GHG budget and avoid stating a full N budget as this could be only based on very rough estimates. We also decided to adjust the text and mention only C and N gains/losses.*

Is there a way to estimate the N content of the fodder/grass on the field before tillage to assess the emissions from the herbage being tilled into the soil?

*We have thought about this too when preparing the manuscript and realized that we had not taken such measurements. However, to our current knowledge the additional N being incorporated into soil during tillage should be very small due the very low vegetation height at this time of the year.*

Does the carbon budget take into account vehicle use? Is it insignificant or does tractor diesel have a role to play?

*The currently presented budget does not include C emissions from vehicle use for two reasons: (1) the hours farm vehicles are being used on this field are very limited over the course of the year given the small size of the fields (negligible). The negligibility of these emissions was further underlined (2) by a MSc thesis that investigated full farmgate budgets in the years prior this study.*

L225: Can you explain what you mean by an internal reference cell in the instrument for the QCLAS? To my knowledge, these cells are used to find absorption lines on the spectra and not for calibration as they leak over time. The QCLAS system typically does not require calibration as it operates on the principles that the absorption follows Hitran quantum mechanics laws.

*Thank you for this comment and this seems to be a misunderstanding of what we have written. We stated that the infrared gas analyser was calibrated regularly, while we also wrote that the QCLAS was fitted against an internal reference cell. In order to create better clarity we changed this sentence as follows: "The QCLAS did not need calibration due to its operating principles, and an internal reference cell (mini-QCL manual, Aerodyne Research Inc., Billerica, MA, USA) eased finding the absorption spectra after each restart of the analyzer."*

Some minor corrections
L283: I think there is a bit of wording here that is confusing. Flushing the chamber with the syringe isn't technically correct. I think it would be better to say that the syringe was used to pump the chamber to circulate the air to avoid the concentration gradients?

*Done*

L471: here the order of the sentences makes it sound like CH4 contributed to 70% of the budget. Please re-order.

*Done, we added "the contribution of $CO_2$"*

L606: Change highlight to highlights

*Done*

[revised manuscript text omitted]

---

## Author Response (AR2)

**Response to reviewers and handling editors comments from 8[th] January 2021 on our manuscript "Memory effects on greenhouse gas emissions (CO2, N2O and CH4) following grassland restoration?" by Lutz Merbold et al.**

We thank both reviewers and the handling editor for the final positive evaluation of our manuscript. We adjusted the manuscript as requested. Similar to the previous exchange, in the following sections, the reviewer/editor's comments are stated first, followed by our response in *italic* font.

**Reviewer #1:**

Having initially reviewed an earlier version of this manuscript, I am pleased to see most of my concerns have been addressed and alleviated (aside from a couple of outstanding points described below). Within this review, I was also asked to comment on the methodological concerns highlighted by the second reviewer, which have not been modified in the current revision of the manuscript. I am of the belief that the manuscript as it currently stands is sufficiently sound methodologically and suitable for publication after addressing a few minor points. Below I have firstly commented on the methodology concerns of the second reviewer, then concluded with a few minor points requiring the authors to address.

Comment on Methodology

The primary concerns of the second reviewer were the use of a running median as a gap-filling method for N2O, and the lack of uncertainties presented. In general, I am in agreement with the author's response on most matters. Given the main purpose of this study was to calculate and present annual GHG exchange from a managed pasture a complete time series of N2O data was a required. As the authors describe, N2O gap-filling is rapidly advancing field with many approaches, but as yet there is no generally accepted and standardised method. A running median approach (particularly for the EC measurements) is as valid as any other method, and certainly appropriate within the scope of this study. Having said that, I find gap-filling of the chamber N2O fluxes with their limited data coverage, (particularly as a single measured is considered representative of a day) rather tenuous, especially given the log-normal distribution of fluxes as noted by the second reviewer. Nonetheless, it is a suitable option to provide an annual estimate of N2O emissions, but this limitation should be commented on within the manuscript where these results are presented.

The lack of uncertainties presented in this study is in line with other recent GHG studies. Certainly, an uncertainty estimate could be applied to the CO2 fluxes where there are well-established methods, and maybe methane, but there are very few papers where uncertainty associated with annual N2O estimates is calculated . Accordingly, given the purpose of this study (annual GHG balances), inclusion of uncertainty to some components and not others prevents an uncertainty estimate for the GHG balance and thus are of marginal benefit. Therefore, I see the results as presented acceptable and in line with current practice.

As mentioned before, we are grateful for the positive evaluation.

Minor Comments and Corrections
• Lines 302-304: I think it would be appropriate to add the % of flux values rejected due to r2 < 0.8 (this would also be in line with the second reviewers comment #3).

We added the link to the paper Imer et a. 2013 where the chamber fluxes of 2010 and 2011 were first presented. In addition, we updated Table 1 – Data availability. This now shows the sampling days as well as the HQ data, i.e. how many data points were above the detection limit for both CH4 and N2O in the years 2010, 2011 and 2013. While there were no changes to N2O, we needed to adjust the number of HQ values for CH4, which does not change the conclusion of the paper but rather further confirm that the 2010 and 2011 values can not be used for GWP calculation.

• Lines 468-470 (and Table 2): for 2010 and 2011 where an annual methane flux is not available, and the annual N2O flux is calculated from very limited data, I question the validity of presenting annual GWP values here for these years. Suggest revising this sentence, and at worst including the disclaimer than 2010 and 2011 are incomplete budgets. (Also, note that the value in line 469 of -387 is now -385 in Table 2).

We added the following sentence in the revised manuscript: As an important note, due to the limited data availability for the years 2010 and 2011, the budgets of these years are likely incomplete. Furthermore, we adjusted Table 2 by similarly highlighting the careful interpretation of the 2010/2011 budgets for N2O.

• Line 565: My initial review highlighted the need to acknowledge the role of enteric methane, which the authors have now done, but I believe there to be a calculation error here. I believe their calculation should be as follows:

1. 404 g CH4 head-1 day-1 * 4.04 head ha-1 * 30 days / 1000 g kg-1 = 48.96 kg CH4 ha-1
2. 48.96 kg CH4 ha-1 * 12/16 = 36.72 kg CH4-C ha-1
3. 36.72 kg CH4-C ha-1 = 3.67 g CH4-C m-2
This is a value is an order of magnitude larger than the 0.407 g CH4-C m-2 stated in the manuscript. This should be corrected, but does not affect the manuscript in any significant manner.

This is correct, thanks for pointing this out. We adjusted this value in the text.

As an aside, the authors could not follow my initial calculations as presented in my first review, and my apologies, I also made a mistake, and they should have been as follows:
1. 1769.9 kg C ha-1 * 3% = 53.1 kg CH4-C ha-1 (proportion of grazed pasture returned as CH4 assuming 3% is returned as methane)
2. 53.1 kg CH4-C ha-1 = 5.31 g CH4-C m-2
This value would be broadly similar to the above calculations by the authors, and thus I find their calculation suitable.

Thank you for clarifying and indeed the magnitude is similar to our estimate.

• Line 597: The start to this paragraph feels like it should be included in the previous paragraph. Revise and correct if needed.

Done

• Table 4: please check the Parcel A and Parcel B labelling for Fertilizer and Harvest lines – these still appear incorrectly labelled.

Corrected

**Comments by Sara Vicca – Handling Editor**

Dear authors,

The two reviewers have now seen your revised manuscript. Despite some disagreement about the methodology, the first reviewer did not provide further comments and indicated that the manuscript could be accepted for publication.
I asked also for the opinion of the other reviewer regarding the methodology and also verified this myself. I agree with the second reviewer that your approach is suitable for the scope of the study.

Thank you for also evaluating the methodology and the positive assessment.

The second reviewer provided some minor suggestions that I would like you to consider before accepting your manuscript for publication. I agree that for 2010 and 2011, the limitations for the GWP calculation need to be clarified in the results section and in Table 2. Note also that the legend of Table 2 was also not fully visible in the uploaded pdf.

We have checked Table 2and adjusted this and similarly we clarified the limitations for GWP calculation in the year 2010 and 2011 in the text. See also our response to the reviewer. We have further corrected